# Model-based Offline RL via Robust Value-Aware Model Learning with Implicitly Differentiable Adaptive Weighting

**Zhongjian Qiao**[1], **Jiafei Lyu**[2], **Boxiang Lyu**[3], **Yao Shu**[4], **Siyang Gao**[1], **Shuang Qiu**[1*]
[1]CityUHK, [2]Tencent, [3]UChicago, [4]HKUST(GZ)
`zhongqiao2-c@my.cityu.edu.hk, shuanqiu@cityu.edu.hk`

## Abstract

Model-based offline reinforcement learning (RL) aims to enhance offline RL with a dynamics model that facilitates policy exploration. However, *model exploitation* could occur due to inevitable model errors, degrading algorithm performance. Adversarial model learning offers a theoretical framework to mitigate model exploitation by solving a maximin formulation. Within such a paradigm, RAMBO (Rigter et al., 2022) has emerged as a representative and most popular method that provides a practical implementation with model gradient. However, we empirically reveal that severe Q-value underestimation and gradient explosion can occur in RAMBO with only slight hyperparameter tuning, suggesting that it tends to be overly conservative and suffers from unstable model updates. To address these issues, we propose **RO**bust value-aware **M**odel learning with **I**mplicitly differentiable adaptive weighting (ROMI). Instead of updating the dynamics model with model gradient, ROMI introduces a novel robust value-aware model learning approach. This approach requires the dynamics model to predict future states with values close to the minimum Q-value within a scale-adjustable state uncertainty set, enabling controllable conservatism and stable model updates. To further improve out-of-distribution (OOD) generalization during multi-step rollouts, we propose implicitly differentiable adaptive weighting, a bi-level optimization scheme that adaptively achieves dynamics- and value-aware model learning. Empirical results on D4RL and NeoRL datasets show that ROMI significantly outperforms RAMBO and achieves competitive or superior performance compared to other state-of-the-art methods on datasets where RAMBO typically underperforms. Code is available at `https://github.com/zq2r/ROMI.git`.

## 1 Introduction

Model-based offline RL (Yu et al., 2020; Kidambi et al., 2020; Rigter et al., 2022; Qiao et al., 2025; 2024) aims to enhance offline RL (Levine et al., 2020; Prudencio et al., 2023) by learning an environmental dynamics model in which the policy could explore. Compared with model-free offline RL (Kumar et al., 2020; Lyu et al., 2022; Jin et al., 2021; Fujimoto & Gu, 2021; Wu et al., 2019; Qiao et al., 2026), model-based offline RL exhibits higher data efficiency and could generalize better to states not present in the dataset. However, model-based offline RL suffers from *model exploitation* (Kidambi et al., 2020), that is, the policy might explore the regions where the model cannot accurately predict the dynamics. Utilizing transitions from these inaccurate regions could degrade the algorithm's performance. Therefore, incorporating conservatism (a.k.a. *pessimism*) is necessary for model-based offline RL to mitigate model exploitation.

One mainstream approach is uncertainty estimation with various heuristics (Yu et al., 2020; Kidambi et al., 2020; Sun et al., 2023; Kim & Oh, 2023). However, uncertainty estimation could be unreliable for neural networks (Rigter et al., 2022; Yu et al., 2021; Lu et al., 2021). Adversarial model learning (Uehara & Sun, 2021) offers another theoretical framework to mitigate model exploitation by solving a maximin formulation, without uncertainty estimation. Among its practical implementations (Rigter et al., 2022; Bhardwaj et al., 2023; Yang et al., 2023), RAMBO (Rigter et al., 2022)

---
*Corresponding Author

has emerged as a representative and widely adopted method, which (1) forces the dynamics model to reduce the value function in OOD regions by minimizing the adversarial loss with *model gradient*, (2) introduces a tradeoff coefficient $\lambda$ to the adversarial term to balance the adversarial training and maximum likelihood estimation (MLE). Although RAMBO provides a practical implementation for adversarial model learning, we observe that the tradeoff coefficient $\lambda$ is set to an extremely small value (e.g., 3e-4) in its implementation, minimizing the effect of the adversarial term. In our further investigation, we empirically find that a slightly larger yet still small $\lambda$ (e.g., 0.05, 0.1) could cause severe Q-value underestimation and even gradient explosion, resulting in training collapse. This indicates that RAMBO **has difficulties in controlling the conservatism level and can be unstable during training.**

To address these issues, we propose **RO**bust value-aware **M**odel learning with **I**mplicitly differentiable adaptive weighting (ROMI). In contrast to RAMBO, ROMI discards the model gradient method that could be overly conservative and unstable, and introduces a novel robust value-aware model learning (Farahmand et al., 2017; Voelcker et al., 2022) approach to incorporate conservatism. Specifically, we require the dynamics model to predict future states with values close to the minimum Q-value within a scale-adjustable state uncertainty set, and adjusting the scale of the uncertainty set is equivalent to controlling the degree of conservatism. This offers flexible and controllable conservatism adjustment and stabilizes model update. However, this value-aware learning scheme solely enforces value awareness and overlooks dynamics awareness that crucially benefits OOD generalization. To close this gap, we further propose implicitly differentiable adaptive weighting, which achieves both dynamics and value awareness in a bi-level optimization framework. Concretely, the inner level incorporates an adaptive weighting network that re-weighs each training sample in an MLE loss, while the outer level updates the weighting network by minimizing the robust value-aware model loss with implicit differentiation. This hierarchical framework adaptively balances dynamics and value-aware model learning. Lastly, we empirically evaluate ROMI on various D4RL (Fu et al., 2020) and NeoRL (Qin et al., 2022) datasets. Our experimental results demonstrate that ROMI significantly surpasses RAMBO, matching or surpassing state-of-the-art (SOTA) methods such as MOBILE (Sun et al., 2023) and Count-MORL (Kim & Oh, 2023), especially on datasets where RAMBO typically underperforms.

## 2 BACKGROUND

**RL.** We consider a Markov Decision Process (MDP) (Puterman, 1990) that can be specified by a tuple $\langle \mathcal{S}, \mathcal{A}, T, r, \rho, \gamma \rangle$, where $\mathcal{S}$ and $\mathcal{A}$ are the state space and action space, respectively, $T : \mathcal{S} \times \mathcal{A} \to \Delta(\mathcal{S})$ is the transition dynamics, and $\Delta(\cdot)$ is the probability simplex. $r : \mathcal{S} \times \mathcal{A} \to \mathbb{R}$ is the reward function, $\rho$ is the initial state distribution, and $\gamma \in [0, 1)$ is the discount factor. The goal of RL is to obtain a policy $\pi_\theta$ which maximizes the objective function: $J(\theta) = \mathbb{E}_{\pi_\theta} \left[ \sum_{t=0}^\infty \gamma^t r(s_t, a_t) | s_0 \sim \rho \right]$.

**Model-based Offline RL.** In offline RL, the agent only has access to a pre-collected offline dataset: $\mathcal{D} = \{(s_i, a_i, r_i, s_{i+1})\}_{i=1}^N$. Model-based offline RL typically trains a dynamics model to augment the dataset. The policy could perform $h$-step rollouts within the dynamics model starting from states in $\mathcal{D}$, and store the collected samples in $\mathcal{D}_{\text{model}}$. However, the dynamics model could deviate from the true dynamics. Therefore, conservatism is necessary to prevent model exploitation.

**Adversarial Model Learning for Model-based Offline RL.** Uehara & Sun (2021) first proposes a theoretical framework that incorporates adversarial model learning into model-based offline RL to introduce conservatism. The offline RL problem is cast as a zero-sum game against an adversarial dynamics model within an uncertainty set, and thus the object is to find a policy $\pi$ from the policy class $\Pi$ by solving the maximin problem:

$$
\pi = \arg\max_{\pi \in \Pi} \min_{\widehat{T} \in \mathcal{M}_\xi} V_{\widehat{T}}^\pi
$$
$$
\text{s.t. } \mathcal{M}_\xi = \left\{ \widehat{T} \mid \mathbb{E}_{(s,a) \sim \mathcal{D}} \left[ \text{Dis}\left( \widehat{T}_{\text{MLE}}(\cdot | s, a), \widehat{T}(\cdot | s, a) \right) \right] \leq \xi \right\},
$$

(1)

where $\mathcal{M}_\xi$ is the dynamics uncertainty set and $\text{Dis}(\cdot, \cdot)$ is the distributional discrepancy measure, $\widehat{T}_{\text{MLE}}$ represents the dataset dynamics learned with maximum likelihood estimation (MLE), i.e, $\widehat{T}_{\text{MLE}} = \arg\max_T \mathbb{E}_{(s,a,s') \in \mathcal{D}} [\log T(s'|s, a)]$. While this offline RL framework offers strong theoretical guarantees, its practical algorithm remains to be explored. A widely-adopted implementation

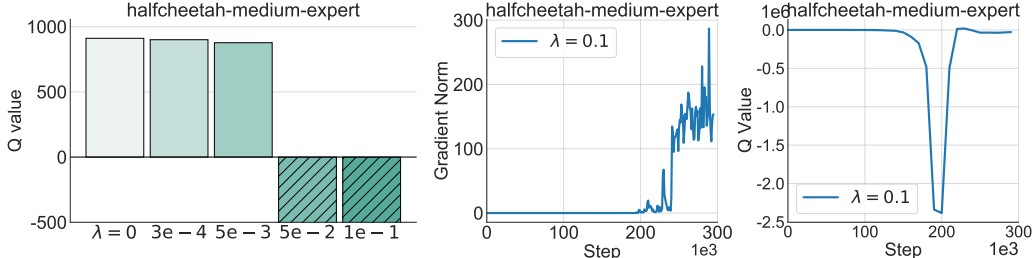

Figure 1: **Left:** Q-value estimation for different values of $\lambda$. The slash-shaded bars indicate that the Q-value has diverged. **Middle:** Gradient norm of the adversarial loss recorded during training. **Right:** Q-value estimation recorded during training.

of this framework is **RAMBO** (Rigter et al., 2022), which optimizes the following objective:

$$\pi = \arg\max_{\pi \in \Pi} \min_{\widehat{T}} \left\{ \lambda V_{\widehat{T}}^\pi - \mathbb{E}_{(s,a,s') \sim \mathcal{D}} \left[ \log \widehat{T}(s'|s,a) \right] \right\}, \tag{2}$$

where the first term is the adversarial loss that corresponds to minimizing the value in OOD regions, optimized via model gradient, and the second term relates to minimizing the negative log-likelihood on the offline data, which is a standard MLE. Here $\lambda$ is treated as a Lagrange multiplier but, for practicality, is hand-tuned and fixed per task.

## 3 MOTIVATION EXAMPLE: THE LIMITATIONS OF RAMBO

In this section, we empirically examine and identify key limitations of RAMBO: **the conservatism degree is hard to control via $\lambda$, and training can be unstable.** Our key observation is that the adversarial weighting coefficient $\lambda$ is set to 3e-4 in the original paper, which is an exceedingly small value for the weighting coefficient. This implies that the effect of the adversarial term should be minimized. We would like to explore the effect of a larger $\lambda$ on training.

We vary $\lambda$ across $\{0, 3\text{e-}4, 5\text{e-}3, 5\text{e-}2, 1\text{e-}1\}$, and run RAMBO on 9 D4RL datasets. We show the results on `halfcheetah-medium-expert-v2` dataset in Figure 1. We find that setting $\lambda$ across $\{0, 3\text{e-}4, 5\text{e-}3\}$ does not show a clear difference in Q-value estimation as illustrated in Figure 1 **Left**, indicating that the adversarial training has little effect on conservatism with a too small $\lambda$. However, when increasing $\lambda$ to larger but still small values within $\{5\text{e-}2, 1\text{e-}1\}$, we observe that the Q-value diverges to a very negative value. This suggests that RAMBO is highly sensitive to the choices of $\lambda$. When $\lambda = 0.1$, we even observe the gradient of the adversarial loss explodes, and the Q-value drops sharply, leading to training collapse, as shown in Figure 1 **Middle**, **Right**. This phenomenon implies that RAMBO tends to be overly conservative and unstable. More details and results on all 9 datasets are presented in Appendix F.2.

We take a deeper analysis of this phenomenon. On the one hand, $\lambda$ is the Lagrange multiplier and should be optimized with gradient ascent (Equation (7) in Rigter et al. (2022)). It is hard to precisely control the degree of conservatism by simply tuning $\lambda$. On the other hand, model gradient would encourage the dynamics model to explore regions with a sharp decrease in value, which may lead to Q-value underestimation and gradient explosion. These factors force $\lambda$ to remain extremely small to minimize the impact of the adversarial term, thereby limiting RAMBO's potential. Therefore, a more practical implementation for Equation 1 that allows flexible adjustment of the conservatism level and enhances training stability is highly desirable.

## 4 METHODOLOGY

Section 3 highlights two major limitations of RAMBO: **(1)** *difficulties in controlling the degree of conservatism*, resulting in severe value underestimation, i.e., over-conservatism; **(2)** *gradient explosion*, leading to unstable training. To address these issues, we develop our method ROMI within the framework of Equation 1. Our central contribution is a novel treatment of adversarial model learning, whose core components, **robust value-aware model learning** and **implicitly differentiable adaptive weighting**, are described in Section 4.1 and Section 4.2 respectively.

### 4.1 ROBUST VALUE-AWARE MODEL LEARNING

Recall that the adversarial model learning paradigm in Equation 1 seeks an adversarial dynamics model for any policy $\pi$ via the inner constrained minimization problem $\min_{\widehat{T}\in\mathcal{M}_\xi} V_{\widehat{T}}^\pi$ with $\mathcal{M}_\xi = \{\widehat{T} \mid \mathbb{E}_{(s,a)\sim\mathcal{D}}[\mathrm{Dis}(\widehat{T}_{\mathrm{MLE}}(\cdot|s,a), \widehat{T}(\cdot|s,a))] \leq \xi\}$. As stated previously in Equation 2, RAMBO explores an implementation of this inner constrained optimization by using a modified Lagrangian-multiplier approach, leading to $\min_{\widehat{T}}\{\lambda V_{\widehat{T}}^\pi - \mathbb{E}_{(s,a,s')\sim\mathcal{D}}[\log\widehat{T}(s'|s,a)]\}$ with a hand-tuned multiplier $\lambda$. This reformulation induces over-conservatism and unstable training, according to our analysis in Section 3.

To tackle such a challenge, we propose an alternative implementation of Equation 1, a robust *value-aware* model learning framework that incorporates conservatism, from a perspective of explicitly characterizing the one-step value estimation error induced by adversarial model learning. Letting $\widehat{T}_m$ be the dynamics model to be learned, we reformulate Equation 1 as follows:

$$\pi = \arg\max_{\pi\in\Pi} V_{\widehat{T}_m}^\pi \quad \text{s.t.} \quad V_{\widehat{T}_m}^\pi = \min_{\widehat{T}\in\mathcal{M}_\xi} V_{\widehat{T}}^\pi, \tag{3}$$

We next focus on the condition $V_{\widehat{T}_m}^\pi = \min_{\widehat{T}\in\mathcal{M}_\xi} V_{\widehat{T}}^\pi$ in Equation 3, which motivates us to minimize a value-aware model error $\mathcal{E}_{\widehat{T}_m}^\pi$ defined in terms of the value difference as follows:

$$\begin{aligned}
\mathcal{E}_{\widehat{T}_m}^\pi &:= \mathbb{E}_{s\sim\rho_\mathcal{D}(\cdot)}\left| V_{\widehat{T}_m}^\pi(s) - \min_{\widehat{T}\in\mathcal{M}_\xi} V_{\widehat{T}}^\pi(s) \right| \\
&= \gamma\cdot\mathbb{E}_{s\sim\rho_\mathcal{D}(\cdot), a\sim\pi(\cdot|s)}\left| \mathbb{E}_{s'\sim\widehat{T}_m(\cdot|s,a)} V_{\widehat{T}_m}^\pi(s') - \min_{\widehat{T}\in\mathcal{M}_\xi}\mathbb{E}_{s'\sim\widehat{T}(\cdot|s,a)} V_{\widehat{T}}^\pi(s') \right|.
\end{aligned} \tag{4}$$

We consider only $s\sim\rho_\mathcal{D}(\cdot)$ here, since the rollout initial states are drawn from the dataset $\mathcal{D}$. Given that $\mathcal{M}_\xi$ is only defined on $\mathcal{D}$ and $a\sim\pi(\cdot|s)$ contains OOD regions, directly minimizing $\mathcal{E}_{\widehat{T}_m}^\pi$ could lead to arbitrarily low Q-value estimates in OOD regions. RAMBO's over-conservatism arises for essentially the same reason. To resolve this issue, we propose to minimize $\widehat{\mathcal{E}}_{\widehat{T}_m}^\pi$ as an alternative:

$$\widehat{\mathcal{E}}_{\widehat{T}_m}^\pi := \gamma\cdot\mathbb{E}_{s\sim\rho_\mathcal{D}(\cdot), a\sim\mu(\cdot|s)}\left| \mathbb{E}_{s'\sim\widehat{T}_m(\cdot|s,a)} \widehat{V}^\pi(s') - \min_{\widehat{T}\in\mathcal{M}_\xi}\mathbb{E}_{s'\sim\widehat{T}(\cdot|s,a)} \widehat{V}^\pi(s') \right|.$$

The blue parts highlight the difference between $\mathcal{E}_{\widehat{T}_m}^\pi$ and $\widehat{\mathcal{E}}_{\widehat{T}_m}^\pi$, where $\mu(\cdot|s)$ is the behavior policy generating $\mathcal{D}$, and $\widehat{V}^\pi$ (abbreviated as $\widehat{V}$ hereafter) represents the value function during training. Intuitively, we learn a mildly conservative value function in the in-distribution (ID) regions, and the conservatism in OOD regions relies on generalization, which we analyse in Proposition 4.2.

The main challenge of minimizing $\widehat{\mathcal{E}}_{\widehat{T}_m}^\pi$ lies in computing $\min_{\widehat{T}\in\mathcal{M}_\xi}\mathbb{E}_{s'\sim\widehat{T}(\cdot|s,a)}\widehat{V}(s')$, since we are not accessible to the dynamics uncertainty set $\mathcal{M}_\xi$, which is crucial for controlling the level of conservatism. To address this, we take the distance measure Dis to be the Wasserstein distance and define the Wasserstein dynamics uncertainty set in Definition 4.1, and exploit its dual form in Proposition 4.1, which provides new insights for handling this issue.

**Definition 4.1** (Wasserstein Dynamics Uncertainty Set). *We define the dynamics uncertainty set via the Wasserstein distance as*

$$\mathcal{M}_\xi = \left\{ \widehat{T} \mid \mathbb{E}_\mathcal{D}\left[ \mathcal{W}\left( \widehat{T}_{\mathrm{MLE}}(\cdot|s,a), \widehat{T}(\cdot|s,a) \right) \right] \leq \xi \right\},$$

*where* $\mathcal{W}\left( \widehat{T}_{\mathrm{MLE}}(\cdot|s,a), \widehat{T}(\cdot|s,a) \right) = \inf_{\zeta\in\Gamma(\widehat{T}_{\mathrm{MLE}}(\cdot|s,a), \widehat{T}(\cdot|s,a))} \mathbb{E}_{(s'_1,s'_2)\sim\zeta}[d(s'_1, s'_2)]$ *is the Wasserstein distance between dynamics* $\widehat{T}_{\mathrm{MLE}}$ *and* $\widehat{T}$, $\Gamma(\cdot,\cdot)$ *is the set of joint distributions whose marginals are* $\widehat{T}_{\mathrm{MLE}}$ *and* $\widehat{T}$, *and* $d(\cdot,\cdot)$ *can be chosen as the Euclidean distance.*

**Proposition 4.1** (Dual Reformulation). *Let* $\mathcal{M}_\xi$ *be the Wasserstein dynamics uncertainty set defined in Definition 4.1. Then we have*

$$\min_{\widehat{T}\in\mathcal{M}_\xi}\mathbb{E}_{s'\sim\widehat{T}(\cdot|s,a)}\widehat{V}(s') = \mathbb{E}_{s'\sim\widehat{T}_{\mathrm{MLE}}(\cdot|s,a)}\left[ \min_{\hat{s}\in U_\xi(s')} \widehat{V}(\hat{s}) \right],$$

*where* $U_\xi(s') = \{\hat{s} \mid d(\hat{s}, s') \leq \xi\}$ *is the state uncertainty set.*

Proposition 4.1 states that, under the Wasserstein distance, the dynamics uncertainty set could be transformed into the state uncertainty set. Based on this reformulation, we rewrite $\widehat{\mathcal{E}}_{\widehat{T}_m}^{\pi}$ as

$$\widehat{\mathcal{E}}_{\widehat{T}_m}^{\pi} = \gamma \cdot \mathbb{E}_{s \sim \rho_{\mathcal{D}}(\cdot), a \sim \mu(\cdot|s)} \left| \mathbb{E}_{s' \sim \widehat{T}_m(\cdot|s,a)} \widehat{V}(s') - \mathbb{E}_{s' \sim \widehat{T}_{\text{MLE}}(\cdot|s,a)} \left[ \min_{\hat{s} \in U_\xi(s')} \widehat{V}(\hat{s}) \right] \right|. \tag{5}$$

To practically minimize $\widehat{\mathcal{E}}_{\widehat{T}_m}^{\pi}$, we further consider an empirical version of $\widehat{\mathcal{E}}_{\widehat{T}_m}^{\pi}$ based on the sample from the offline data, i.e., $(s, a, s') \sim \mathcal{D}$, in place of $s \sim \rho_{\mathcal{D}}(\cdot), a \sim \mu(\cdot|s), s' \sim \widehat{T}_{\text{MLE}}(\cdot|s,a)$, since $\mathcal{D}$ was collected through the same sampling process. Thus, given that $s'$ is present in $\mathcal{D}$, we can simply construct $U_\xi(s')$ by perturbing $s'$ with noise of scale $\xi$ and performing random sampling. We further parameterize the dynamics model $\widehat{T}_m$ by $\psi$, denoted by $\widehat{T}_\psi$. Eventually, Equation 5 is reformulated as a **R**obust **V**alue-aware model **L**earning (RVL) loss:

$$\mathcal{L}_{\text{RVL}}(\psi) := \mathbb{E}_{(s,a,s') \in \mathcal{D}} \left( \mathbb{E}_{\hat{s}' \sim \widehat{T}_\psi(\cdot|s,a)} \widehat{V}(\hat{s}') - \min_{\{\tilde{s}'_i\}_{i=1}^N \in U_\xi(s')} \widehat{V}(\tilde{s}'_i) \right)^2, \tag{6}$$

where $\widehat{V}$ is chosen as the target value network for training stability, $\{\tilde{s}'_i\}_{i=1}^N$ are sampled from $U_\xi(s')$, and $N$ is the sample size. We note that Equation 6 differs from the typical value-aware model loss (Farahmand et al., 2017; Voelcker et al., 2022). Our approach contributes to incorporating conservatism via a state uncertainty set into the loss for offline RL. Specifically, Equation 6 requires the dynamics model to predict the future state whose value is close to the minimum value for states within the state uncertainty set. **Moreover, the scale $\xi$ of the uncertainty set precisely quantifies the level of conservatism** according to Proposition 4.1. Thus, we could simply adjust the uncertainty set scale $\xi$ to achieve controllable conservatism and enhance training stability.

Note that Equation 6 only utilizes the dataset samples, and does not explicitly regularize the values for the OOD region that may be encountered during policy rollouts. To address this concern, we first analyze the impact of potential generalization errors on training. We define the maximum generalization error of Equation 6 during rollouts as

$$\epsilon_1 := \max_{(s,a) \in \mathcal{D}_{\text{model}}} \left| \mathbb{E}_{\hat{s}' \sim \widehat{T}_\psi(\cdot|s,a)} \left[ \widehat{V}(\hat{s}') \right] - \mathbb{E}_{s' \sim \widehat{T}_{\text{MLE}}(\cdot|s,a)} \left[ \min_{\tilde{s}' \in U_\xi(s')} \widehat{V}(\tilde{s}') \right] \right|,$$

where $\mathcal{D}_{\text{model}}$ denotes the dataset consisting of synthetic transition trajectories generated via policy rollouts, within which OOD data can reside. We also define the maximum value discrepancy between true next states and those within the state uncertainty set $U_\xi$ as

$$\epsilon_2 := \max_{(s,a) \in \mathcal{D}_{\text{model}}} \left| \mathbb{E}_{s' \sim T(\cdot|s,a)} \left[ \widehat{V}(s') \right] - \mathbb{E}_{s' \sim \widehat{T}_{\text{MLE}}(\cdot|s,a)} \left[ \min_{\tilde{s}' \in U_\xi(s')} \widehat{V}(\tilde{s}') \right] \right|.$$

Then the learned Q-function by minimizing Equation 6 maintains a bounded value during policy rollouts, as guaranteed by Proposition 4.2.

**Proposition 4.2** (Bounded Q-value). *The learned Q-function $\widehat{Q}$ satisfies:*

$$Q_{\text{true}}(s,a) - \frac{\gamma(\epsilon_1 + \epsilon_2)}{1 - \gamma} \leq \widehat{Q}(s,a) \leq Q_{\text{true}}(s,a) + \frac{\gamma \epsilon_1}{1 - \gamma}, \qquad \forall (s,a) \in \mathcal{D}_{\text{model}}, \tag{7}$$

*where $Q_{\text{true}}$ denotes the true Q-function.*

Proposition 4.2 indicates that the learned Q-value would stay bounded if $\epsilon_1$ and $\epsilon_2$ are bounded. $\epsilon_2$ is monotonically increasing w.r.t. $\xi$, indicating that we can bound $\epsilon_2$ by adjusting $\xi$. We demonstrate how we minimize the impact of $\epsilon_1$ in Section 4.2.

## 4.2 IMPLICITLY DIFFERENTIABLE ADAPTIVE WEIGHTING

We note that the generalization error $\epsilon_1$ could be large during multi-step rollouts, since $\mathcal{L}_{\text{RVL}}$ only focuses on value awareness while ignoring dynamics awareness, which could be crucial for OOD generalization. To demonstrate this, let $s_0$ be the rollout starting point, and $a_0 \sim \pi(\cdot|s_0)$ be the policy output. The dynamics model needs to predict the next state $s_1$. Since $\mathcal{L}_{\text{RVL}}$ only regularizes the value of $s_1$, the predicted $s_1$ may deviate significantly from the true dynamics transition. In subsequent steps, the policy will be forced to explore OOD regions, and the dynamics model will

also need to predict transitions in these OOD regions. This may lead to compounding prediction errors and a large $\epsilon_1$. Therefore, **how can we achieve value awareness to ensure conservatism while improving dynamics awareness for better generalization?**

Inspired by previous works (Shu et al., 2019; Jiang et al., 2022; Yuan et al., 2023), we propose to optimize Equation 6 with implicitly differentiable adaptive weighting, which integrates dynamics- and value-aware model learning in a bi-level optimization framework. Specifically, we introduce an adaptive-weighting network $w_\nu(s, a, s')$ to assign an individual weight to each transition $(s, a, s')$ and optimize the following objective:

$$\min_\nu \mathcal{L}_{\text{RVL}}(\psi(\nu)),$$

$$s.t. \quad \psi^\star(\nu) = \arg\min_\psi \left\{ \mathcal{L}_{\text{WSL}}(\psi, \nu) := \mathbb{E}_{(s,a,s')\in\mathcal{D}} \left[ w_\nu(s, a, s') \log\left( \widehat{T}_\psi(s'|s, a) \right) \right] \right\}, \quad (8)$$

where $\mathcal{L}_{\text{WSL}}$ denotes the Weighted Supervised Learning (WSL) loss, compared to the standard supervised learning loss $\mathcal{L}_{\text{SL}} := \mathbb{E}_{(s,a,s')\in\mathcal{D}} \left[ \log\left( \widehat{T}_\psi(s'|s, a) \right) \right]$. Specifically, we update the dynamics model through weighted supervised learning in the inner level to achieve dynamics awareness, and optimize $w_\nu$ in the outer level by minimizing $\mathcal{L}_{\text{RVL}}$ with implicit differentiation to achieve value awareness.

**Inner Level: Optimizing $\psi$ to Improve Dynamics Awareness.** In the inner level, we fix $w_\nu$ and update the dynamics model by minimizing $\mathcal{L}_{\text{WSL}}$ with typical gradient descent:

$$\psi^{(t+1)} \leftarrow \psi^{(t)} - \beta_{1(t)} \cdot \nabla_\psi \mathcal{L}_{\text{WSL}}(\psi^{(t)}, \nu^{(t)}), \quad (9)$$

where $\beta_{1(t)}$ represents the $t$-step learning rate of the dynamics model.

**Outer Level: Optimizing $\nu$ to Achieve Value Awareness.** The outer level objective is formalized in Equation 6. Since we have established the analytical relationship between $\psi$ and $\nu$ through the inner level, we can implicitly calculate the gradient of $\mathcal{L}_{\text{RVL}}(\psi(\nu))$ w.r.t. $\nu$ using the chain rule:

$$g_{\text{RVL}}^{(t)} = \nabla_\psi \mathcal{L}_{\text{RVL}}(\psi^{(t+1)}(\nu^{(t)}))^\mathsf{T} \cdot \nabla_\nu \psi^{(t+1)}(\nu^{(t)}) = h \cdot \nabla_\nu w_{\nu^{(t)}}(s, a, s'),$$

where $h = -\beta_{1(t)} \cdot \nabla_\psi \mathcal{L}_{\text{RVL}}(\psi^{(t+1)}(\nu^{(t)}))^\mathsf{T} \cdot \nabla_\psi \log\left( \widehat{T}_{\psi^{(t)}}(s'|s, a) \right)$. The derivation is deferred in Appendix A. Then we can update $\nu$ using automatic differentiation in Pytorch (Paszke et al., 2019):

$$\nu^{(t+1)} \leftarrow \nu^{(t)} - \beta_{2(t)} \cdot g_{\text{RVL}}^{(t)},$$

where $\beta_{2(t)}$ is the $t$-step learning rate of the weighting network. Through this bi-level optimization process, the weighting network learns to prioritize samples that contribute most to minimizing $\mathcal{L}_{\text{RVL}}$, and the dynamics model still learns to reconstruct the environmental dynamics, achieving both dynamics and value awareness. Assume (1) $\widehat{V}$ is $L_V$-Lipschitz continuous, that is, $|\widehat{V}(s_1) - \widehat{V}(s_2)| \le L_V \|s_1 - s_2\|$ holds for $\forall s_1, s_2 \in \mathcal{S}$; (2) the dynamics model prediction error during rollouts is bounded by a constant $\epsilon$: $\max_{(s,a)\in\mathcal{D}_{\text{model}}} \mathbb{E}_{\hat{s}'\sim\widehat{T}_\psi(\cdot|s,a), s'\sim\widehat{T}_{\text{MLE}}(\cdot|s,a)} \|\hat{s}' - s'\| \le \epsilon$, which could be satisfied through bi-level optimization and short-branch rollouts (Janner et al., 2019). Then the generalization error $\epsilon_1$ could be bounded according to Corollary 4.1.

**Corollary 4.1** (Bounded $\epsilon_1$). *Under the assumptions of Lipschitz continuity and bounded dynamics model prediction error, the generalization error $\epsilon_1$ could be bounded by*

$$\epsilon_1 \le L_V \cdot (\epsilon + \xi),$$

where $\xi$ is the uncertainty set scale. We also provide the convergence rate analysis for this bi-level optimization framework in Proposition 4.3.

**Proposition 4.3** (Convergence Analysis). *Assume $\mathcal{L}_{\text{RVL}}$ and $\mathcal{L}_{\text{WSL}}$ are $L$-Lipschitz smooth, and the gradient of $\mathcal{L}_{\text{RVL}}$, $\mathcal{L}_{\text{SL}}$ and $\mathcal{L}_{\text{WSL}}$ are bounded by $\rho$. Let $w_\nu$ be twice differentiable, with its gradient and Hessian bounded by $\delta$ and $\mathcal{B}$, respectively. Denote $L' = \rho^2(L\delta^2 + \mathcal{B})$, and the total steps as $K$. For some $c_1, c_2 > 0$, we assume the learning rate of the inner and outer loop $\beta_1$ and $\beta_2$ satisfies: $\frac{c_2}{\sqrt{K}} \le \beta_1, \beta_2 \le \min\{\frac{c_1}{K}, \frac{1}{L}, \frac{1}{L'}\}$, where $\frac{c_2}{\sqrt{K}} \le \min\{\frac{c_1}{K}, \frac{1}{L}, \frac{1}{L'}\}, \frac{c_2}{\sqrt{K}} < 1, \frac{c_1}{K} < 1$. Then the outer loss $\mathcal{L}_{\text{RVL}}$ and inner loss $\mathcal{L}_{\text{WSL}}$ can achieve a convergence rate of $\mathcal{O}(\frac{1}{\sqrt{K}})$, i.e.,*

$$\min_{0 \le t \le K-1} \mathbb{E} \left\| \nabla_\nu \mathcal{L}_{\text{RVL}}(\psi^{(t+1)}(\nu^{(t)})) \right\|^2 \le \mathcal{O}\left( \frac{1}{\sqrt{K}} \right),$$

$$\min_{0 \le t \le K-1} \mathbb{E} \left\| \nabla_\psi \mathcal{L}_{\text{WSL}}(\psi^{(t)}, \nu^{(t)}) \right\|^2 \le \mathcal{O}\left( \frac{1}{\sqrt{K}} \right).$$

Table 1: Normalized score comparison between different offline RL algorithms and ROMI on 12 D4RL MuJoCo datasets. We abbreviate "random" as "r", "medium" as "m", "medium-replay" as "mr", "medium-expert" as "me". Each method is run with 1M steps.

| Task Name | CQL | IQL | MOPO | RAMBO | COUNT | MOBILE[1] | ROMI (Ours) |
|---|---|---|---|---|---|---|---|
| hopper-r | 7.9 | 7.6 | 27.8 | 23.9 | **30.0** | 28.5 | **31.7 ± 0.3** |
| halfcheetah-r | 17.5 | 13.1 | 36.7 | 29.5 | 37.3 | **39.4** | 38.4±4.1 |
| walker2d-r | 5.1 | 5.4 | 0.0 | 0.0 | 19.9 | **23.6** | 22.4 ±2.6 |
| hopper-m | 53.0 | 66.2 | 72.2 | 95.2 | 100.2 | 77.6 | **105.0±3.3** |
| halfcheetah-m | 47.0 | 47.9 | 64.0 | **76.2** | 75.9 | 74.6 | **76.2±0.6** |
| walker2d-m | 73.4 | 67.2 | 82.1 | 83.9 | 88.4 | **90.7** | **93.6±1.3** |
| hopper-mr | 88.7 | 94.0 | 90.5 | 77.2 | 97.6 | 85.7 | **102.0±2.4** |
| halfcheetah-mr | 45.3 | 44.3 | 52.1 | **69.7** | 70.5 | 67.7 | **70.7±1.8** |
| walker2d-mr | 81.8 | 74.8 | 73.6 | 81.2 | **87.5** | 75.0 | 85.2±3.6 |
| hopper-me | 105.9 | 92.7 | 84.5 | 99.7 | **109.8** | 95.6 | **110.5±6.8** |
| halfcheetah-me | 75.6 | 86.3 | 92.3 | 93.9 | 100.0 | 96.1 | **104.5 ±2.2** |
| walker2d-me | 107.9 | **110.7** | 93.8 | 73.7 | **110.4** | 103.2 | **113.3±1.9** |
| Total | 709.1 | 710.2 | 769.6 | 804.1 | 927.5 | 857.7 | **953.5** |

**ROMI Algorithm.** Eventually, combining the dynamics model learning in Equation 8 and policy learning $\pi = \arg\max_{\pi \in \Pi} V_{\widehat{T}_\psi}^\pi$ yields the optimization problem for ROMI. The key distinction from RAMBO lies in the training strategy of the dynamics model, as described in Section 4.1 and Section 4.2. The policy is learned with SAC (Haarnoja et al., 2018) by using the data from $\mathcal{D}$ and $\mathcal{D}_{\text{model}}$. More implementation details of ROMI are presented in Appendix D.2.

## 5 EXPERIMENTS

In this section, we empirically evaluate ROMI. We compare the performance between ROMI and other baselines on standard benchmarks in Section 5.1. We conduct an ablation study to examine the effect of adaptive weighting in Section 5.2, and we verify that ROMI offers flexible conservatism control and avoids gradient explosion across a wide range of $\xi$ in Section 5.3.

### 5.1 BENCHMARK RESULTS

**D4RL.** We compare ROMI with several offline RL algorithms, including model-free methods CQL (Kumar et al., 2020) and IQL (Kostrikov et al., 2021); and model-based methods: MOPO (Yu et al., 2020), RAMBO (Rigter et al., 2022), Count-MORL (Kim & Oh, 2023), and MOBILE (Sun et al., 2023).

For the MuJoCo (Todorov et al., 2012) domain, we consider three tasks (`halfcheetah`, `hopper`, `walker2d`) with four types of datasets (`random`, `medium`, `medium-replay`, `medium-expert`) for each task. The datasets we use are of the "v2" version. Experiments are run with 5 seeds. More evaluations on Antmaze tasks are deferred to Appendix F.1 due to space limitations.

In Table 1, we summarize the evaluation results of ROMI and the compared baselines. We can see that ROMI outperforms RAMBO on **11** out of 12 datasets, especially on datasets where RAMBO underperforms, such as `hopper-medium-replay` and `walker2d-medium-expert`. ROMI achieves a total score of **953.5**, which is **18.6%** higher than that of RAMBO. Furthermore, even when compared to other algorithms, including MOBILE and Count-MORL, ROMI demonstrates superior performance, achieving (one of) the best results on **11** out of 12 datasets. On the remaining datasets, ROMI also delivers performance comparable to the best-performing methods.

---

[1]3M steps are used in MOBILE paper, and we only report the results at 1M steps for a fair comparison.

Table 2: Normalized score comparison between different offline RL algorithms and our method ROMI on NeoRL tasks. Each method is run with 1M steps over 5 seeds.

| Task Name | CQL | IQL | EDAC | MOPO | RAMBO | MOBILE | ROMI (Ours) |
|---|---|---|---|---|---|---|---|
| Hopper-L | 16.8 | 14.3 | 12.2 | 12.0 | 15.1 | 18.1 | **22.4±0.5** |
| HalfCheetah-L | 35.0 | 34.1 | 31.3 | 34.2 | 30.1 | **38.9** | 35.8±3.9 |
| Walker2d-L | **42.4** | 40.1 | **43.1** | 15.6 | 24.3 | 39.0 | 36.4±2.3 |
| Hopper-M | **58.1** | **56.8** | 37.2 | 0.0 | 37.5 | 47.3 | 46.6± 6.8 |
| HalfCheetah-M | 50.0 | 52.0 | 54.6 | 53.1 | 51.2 | **55.6** | 57.7±1.4 |
| Walker2d-M | 52.5 | 51.2 | **53.9** | 35.7 | 36.2 | 51.4 | **54.9±2.0** |
| Hopper-H | **65.6** | 62.4 | 56.7 | 17.9 | 48.7 | **65.0** | 65.9±4.5 |
| HalfCheetah-H | 74.6 | **76.3** | 72.0 | 63.2 | 72.3 | 69.4 | 77.4±2.7 |
| Walker2d-H | 71.3 | **74.0** | 73.1 | 26.4 | 67.4 | 71.7 | 75.1±1.8 |
| Total | 466.3 | 461.2 | 434.1 | 258.1 | 382.8 | 456.4 | **472.2** |

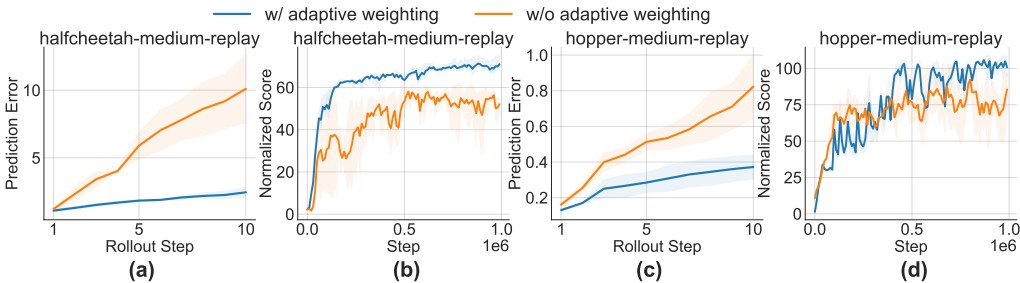

Figure 2: Comparison of the prediction error and performance w/ and w/o adaptive weighting.

**NeoRL.** We also test our method on NeoRL (Qin et al., 2022) benchmark, which is often used to evaluate model-based offline RL (Sun et al., 2023; Lin et al., 2024). We consider nine datasets, which involves three tasks (`halfcheetah`, `hopper`, `walker2d`) and three types of data qualities (`low`, `medium`, `high`). For each dataset, we select 1,000 trajectories uniformly for our experiments. We choose model-free methods (CQL, IQL, EDAC (An et al., 2021)) and model-based methods (MOPO, RAMBO, MOBILE) as compared baselines. We exclude Count-MORL since there are no reference hyperparameter settings available in the original paper or the NeoRL paper.

We summarize the normalized score comparison results in Table 2. We observe that model-free methods outperform model-based baselines on most datasets, while ROMI surpasses all baselines on **6** out of 9 datasets, and also achieves the highest total score. Notably, ROMI outperforms RAMBO on all 9 datasets, verifying the effectiveness of ROMI.

## 5.2 STUDY ON THE EFFECT OF DYNAMICS AWARENESS

In this section, we conduct an ablation study to test the effect of dynamics awareness on the performance and OOD generalization. Specifically, we solely apply Equation 6 for model updates and drop the adaptive weighting component in Section 4.2 to examine its effect on the performance and prediction error during multi-step rollouts. We conduct experiments on `halfcheetah-medium-replay` and `hopper-medium-replay` datasets.

We present the experimental results in Figure 2, where panels (a) and (c) show the model prediction error under different rollout steps with and without adaptive weighting, while panels (b) and (d) display the corresponding learning curves. The results clearly demonstrate that adaptive weighting leads to improved performance and reduced prediction error across various rollout steps. This indicates that the dynamics awareness is crucial for OOD generalization and achieving strong performance.

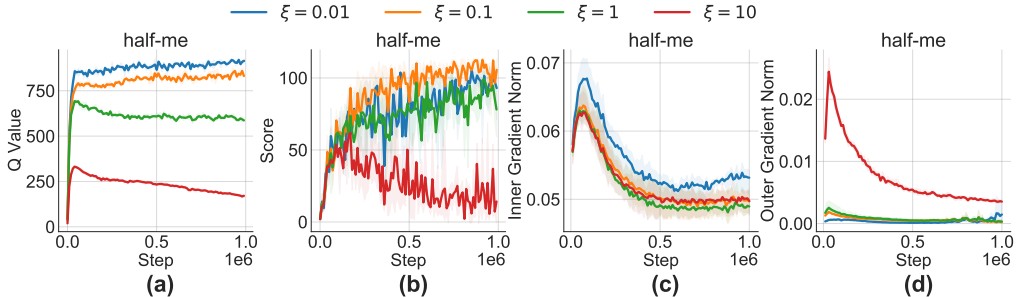

Figure 3: Comparison for different $\xi$ in terms of Q-value, normalized score, inner loss gradient norm, outer loss gradient norm.

### 5.3 STUDY ON THE EFFECT OF $\xi$

In this section, we examine the effect of the key hyperparameter: the uncertainty set scale $\xi$, on training. Specifically, we conduct experiments on `halfcheetah-medium-expert-v2` dataset and vary the value of $\xi$ across a wide range of values: $\{0.01, 0.1, 1.0, 10\}$. A larger value of $\xi$ corresponds to a higher degree of conservatism. We record four metrics during training: the estimated Q-value, normalized score, gradient norm of the inner loss, and gradient norm of the outer loss.

The experimental results are presented in Figure 3. Compared to the results of RAMBO in Section 3, we observe that: (1) neither severe Q-value underestimation nor training collapse occurs even for $\xi = 10$, and the gradient norm stays small throughout training; (2) the Q-values are clearly distinguishable for different $\xi$, with a larger $\xi$ corresponding to a lower Q-value. These verify that ROMI could provide a more controllable degree of conservatism while ensuring training stability.

## 6 RELATED WORK

**Model-based Offline RL.** The core issue of model-based offline RL is how to incorporate conservatism into model learning or policy update to prevent model exploitation. One line of work is to adopt uncertainty estimation. MOPO (Yu et al., 2020) and MoReL (Kidambi et al., 2020) use the uncertainty of model prediction to penalize the reward function to achieve a pessimistic value estimation; SUMO (Qiao et al., 2025) utilizes a KNN-based entropy estimator for uncertainty quantification; Count-MORL (Kim & Oh, 2023) utilizes the count estimates of state-action pairs to measure the uncertainty; MOBILE (Sun et al., 2023) proposes Model-Bellman inconsistency to improve the uncertainty quantification. COMBO (Yu et al., 2021) and RAMBO (Rigter et al., 2022) do not rely on uncertainty estimation and minimize the value estimation on model-generated samples.

**Robust RL.** Similar to RAMBO, ROMI is related to Robust RL. Robust RL optimizes the policy in a Robust MDP (Bagnell et al., 2001; Cohen & Hutter, 2020; Nilim & El Ghaoui, 2005; Wiesemann et al., 2013; Rigter et al., 2021), that is, finding the policy with the best worst-case performance over a set of possible MDPs. Typically, the uncertainty set of MDPs is specified a priori. In contrast, our problem setting differs in that the uncertainty set of MDPs is assumed to be unknown. We need to learn a dynamics model to simulate the worst-case MDP.

**Bi-level Optimization in RL.** Bi-level optimization is a classic framework for optimizing hierarchical learning objectives. There are several studies leveraging bi-level optimization to solve different RL tasks. CAIL (Zhang et al., 2021) re-weights demonstrations with different optimality in imitation learning; Meta-Reward-Net (Liu et al., 2022) incorporates the performance of the Q function into reward learning for preference-based RL; VACO (Jiang et al., 2025) incorporates bi-level optimization in offline RL to balance policy improvement and behavior cloning; TEMPO (Yuan et al., 2023) is closest to our work, which leverages both task-specific and semantic information for effective world model learning. But our method is different since we focus on offline RL and have to consider conservatism in the outer level. There are also recent studies (Li et al., 2024; Chen et al., 2025) incorporating bi-level optimization into Large Language Model (LLM)-oriented RL.

## 7 CONCLUSION

In this paper, we first identify the limitations of RAMBO. We empirically find that RAMBO tends to be overly conservative and incurs instability during training. To address these issues, we propose our method ROMI, which adopts a novel robust value-aware model learning scheme by requiring the dynamics model to predict future states with values close to the minimum value in the state uncertainty set. To further improve OOD generalization ability, we introduce a bi-level optimization framework called implicitly differentiable adaptive weighting, which achieves dynamics and value awareness. Compared to RAMBO, ROMI provides more controllable conservatism and stabilizes training. Empirical Results on D4RL and NeoRL datasets demonstrate the effectiveness of ROMI.

**Limitations.** One limitation of ROMI is that additional computational cost for bi-level optimization is required compared to the original RAMBO. Another limitation is that the conservatism degree governed by $\xi$ has to be specified before training, unlike methods that support runtime adjustment (Ghosh et al., 2022; Hong et al., 2022; Swazinna et al., 2022; Wang et al., 2023). Addressing these limitations presents an interesting direction for future work.

## 8 ACKNOWLEDGEMENTS

This research was supported in part by the Hong Kong Research Grants Council (GRF 11217925, GRF 16209124) and the National Science Foundation of China (Grant 72371214). The authors would also like to thank the anonymous reviewers for their valuable comments on our manuscript.

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

## A    DERIVATION OF THE OUTER LOSS GRADIENT

In Section 4.2, we calculate the gradient of the outer loss at step $t$ by:

$$g_{\mathrm{RVL}}^{(t)} = \nabla_\psi \mathcal{L}_{\mathrm{RVL}}(\psi^{(t+1)}(\nu^{(t)}))^\mathsf{T} \cdot \nabla_\nu \psi^{(t+1)}(\nu^{(t)}) = h \cdot \nabla_\nu w_{\nu^{(t)}}(s, a, s'),$$

where

$$h = -\beta_{1(t)} \nabla_\psi \mathcal{L}_{\mathrm{RVL}}(\psi^{(t+1)}(\nu^{(t)}))^\mathsf{T} \cdot \nabla_\psi \log\left(\widehat{T}_{\psi^{(t)}}(s'|s, a)\right). \tag{10}$$

We provide the complete derivation in this part. We denote the initial value of $\psi$ and $\nu$ as $\psi^{(0)}$ and $\nu^{(0)}$, and make the following stipulations for the outer loop and inner loop updates: **(1)** at each update step $t$, the inner loop is updated first, followed by the outer loop; **(2)** at each update step $t$, only one step of gradient descent is performed. Under above conditions, at step $t$, the inner loop first takes $\psi^{(t)}$ and $\nu^{(t)}$ as input and performs one-step update to obtain $\psi^{(t+1)}$. Then the outer loop takes $\psi^{(t+1)}$ as input, which is a function of $\nu^{(t)}$, and performs implicit update to obtain $\nu^{(t+1)}$. Then we can derive the gradient as follows:

$$
\begin{aligned}
g_{\mathrm{RVL}}^{(t)} &= \nabla_\nu \mathcal{L}_{\mathrm{RVL}}(\psi^{(t+1)}(\nu^{(t)})) \\
&= \nabla_\psi \mathcal{L}_{\mathrm{RVL}}(\psi^{(t+1)}(\nu^{(t)}))^\mathsf{T} \cdot \nabla_\nu \psi^{(t+1)}(\nu^{(t)}) \\
&= \nabla_\psi \mathcal{L}_{\mathrm{RVL}}(\psi^{(t+1)}(\nu^{(t)}))^\mathsf{T} \cdot \nabla_\nu \left(\psi^{(t}(\nu^{(t)}) - \beta_{1(t)} \nabla_\psi \mathcal{L}_{\mathrm{WSL}}(\psi^{(t)}, \nu^{(t)})\right) \\
&= -\beta_{1(t)} \nabla_\psi \mathcal{L}_{\mathrm{RVL}}(\psi^{(t+1)}(\nu^{(t)}))^\mathsf{T} \cdot \nabla_\nu \left(\nabla_\psi \mathcal{L}_{\mathrm{WSL}}(\psi^{(t)}, \nu^{(t)})\right) \\
&= \underbrace{-\beta_{1(t)} \nabla_\psi \mathcal{L}_{\mathrm{RVL}}(\psi^{(t+1)}(\nu^{(t)}))^\mathsf{T} \cdot \nabla_\psi \log\left(\widehat{T}_{\psi^{(t)}}(s'|s, a)\right)}_{h} \cdot \nabla_\nu w_{\nu^{(t)}}(s, a, s').
\end{aligned}
$$

The fourth equality holds under the Markov assumption, under which $\nabla_\nu \psi^{(t)}(\nu^{(t)}) = 0$ since $\psi^{(t)}$ is determined by $\nu^{(t-1)}$ and the effect of $\nu^{(t)}$ is ignored. This technique is widely used in the field of bi-level optimization (Bolte et al., 2023; Liu et al., 2020) and meta-learning (Baydin et al., 2017; Wang et al., 2020). Then we finish our derivation.

## B    MISSING PROOFS

In this section, we provide detailed proofs for our theoretical results in the main text.

### B.1    PROOF OF PROPOSITION 4.1

We first introduce the following lemma in preparation for proving Proposition 4.1.

**Lemma B.1.** *Let $\mathcal{S}$ be a measure space and $P$ be a probability measure on $\mathcal{S}$. In addition, let $f : \mathcal{S} \to \mathbb{R}$ be any measure function and $c : \mathcal{S} \times \mathcal{S} \to \mathbb{R}_{\geq 0}$ be a cost function. Then for any scalar $\lambda \geq 0$, the following equality holds:*

$$\inf_{\hat{P} \sim \mathcal{P}(\mathcal{S})} \left(\mathbb{E}_{\hat{s} \sim \hat{P}}[f(\hat{s})] + \lambda \mathcal{W}(\hat{P}, P)\right) = \mathbb{E}_{s \sim P}\left[\inf_{\hat{s} \sim \mathcal{S}} \left(f(\hat{s}) + \lambda c(s, \hat{s})\right)\right],$$

*where $\mathcal{P}(\mathcal{S})$ represents all probability measures on $\mathcal{S}$, and $\mathcal{W}$ is the Wasserstein distance w.r.t. the cost function $c$.*

*Proof.* We prove this lemma by performing a series of transformations on the left-hand side (LHS) and proving its equivalence to the right-hand side (RHS).

According to the definition of Wasserstein distance, the LHS can be written as

$$\mathrm{LHS} = \inf_{\hat{P} \in \mathcal{P}(\mathcal{S})} \left(\mathbb{E}_{\hat{s} \sim \hat{P}}[f(\hat{s})] + \lambda \inf_{\gamma \in \Gamma(P, \hat{P})} \mathbb{E}_{(s, \hat{s}) \sim \gamma}[c(s, \hat{s})]\right).$$

The optimization of $\hat{P}$ and the inner optimization over $\gamma \in \Gamma(P, \hat{P})$ could be combined into a single optimization over all couplings $\gamma$ whose first marginal is $P$, and the second marginal, $\hat{P}$, could be arbitrary in $\mathcal{P}(\mathcal{S})$. Then we have

$$
\begin{aligned}
\text{LHS} &= \inf_{\gamma \in \Gamma(P,\hat{P})} \left( \mathbb{E}_{\hat{s} \sim \hat{P}}[f(\hat{s})] + \lambda \mathbb{E}_{(s,\hat{s}) \sim \gamma}[c(s,\hat{s})] \right) \\
&= \inf_{\gamma \in \Gamma(P,\hat{P})} \mathbb{E}_{(s,\hat{s}) \sim \gamma} \left[ f(\hat{s}) + \lambda c(s,\hat{s}) \right],
\end{aligned}
\tag{11}
$$

where the second equality holds by the linearity of expectation.

By the disintegration theorem for measures, any coupling $\gamma \in \Gamma(P, \hat{P})$ could be represented as the product of its first marginal $P$ and a stochastic kernel $K(d\hat{s}|s) : \mathcal{S} \to \mathcal{P}(\mathcal{S})$:

$$
\gamma(ds, d\hat{s}) = K(d\hat{s}|s)P(ds).
\tag{12}
$$

This implies that optimizing over all couplings $\gamma \in \Gamma(P, \hat{P})$ is equivalent to optimizing over all possible stochastic kernels $K$. We substitute Equation 12 into Equation 11:

$$
\begin{aligned}
\text{LHS} &= \inf_K \int_{\mathcal{S}} \int_{\mathcal{S}} [f(\hat{s}) + \lambda c(s,\hat{s})] \, K(d\hat{s}|s) P(ds) \\
&= \inf_K \mathbb{E}_{s \sim P} \left[ \mathbb{E}_{\hat{s} \sim K(\cdot|s)} [f(\hat{s}) + \lambda c(s,\hat{s})] \right].
\end{aligned}
$$

We change the position of the infimum operator and the inner expectation:

$$
\text{LHS} = \mathbb{E}_{s \sim P} \left[ \inf_{K(\cdot|s) \in \mathcal{P}(\mathcal{S})} \mathbb{E}_{\hat{s} \sim K(\cdot|s)} [f(\hat{s}) + \gamma c(s,\hat{s})] \right].
\tag{13}
$$

We then solve the inner minimization problem for a fixed $s \in \mathcal{S}$:

$$
\inf_{K(\cdot|s) \in \mathcal{P}(\mathcal{S})} \mathbb{E}_{\hat{s} \sim K(\cdot|s)} [f(\hat{s}) + \gamma c(s,\hat{s})].
$$

Let $g_s(\hat{s}) \triangleq f(\hat{s}) + \gamma c(s,\hat{s})$. The problem is to find a probability measure $K(\cdot|s)$ that minimizes the expectation of $g_s(\hat{s})$. It is obvious that this minimum is achieved by concentrating the entire probability mass on point $\hat{s}$ where $g_s(\hat{s})$ attains its infimum.

Let $\hat{s}^\star = \arg\inf_{\hat{s} \in \mathcal{S}} g_s(\hat{s})$. The optimal measure is a Dirac measure $\delta_{\hat{s}^\star}$ centered on $\hat{s}^\star$. Therefore,

$$
\begin{aligned}
\inf_{K(\cdot|s) \in \mathcal{P}(\mathcal{S})} &\mathbb{E}_{\hat{s} \sim K(\cdot|s)} [f(\hat{s}) + \gamma c(s,\hat{s})] \\
&= \mathbb{E}_{\hat{s} \sim \delta_{\hat{s}^\star}} [g_s(\hat{s})] \\
&= f(\hat{s}^\star) + \gamma c(s,\hat{s}^\star) \\
&= \inf_{\hat{s} \in \mathcal{S}} [f(\hat{s}) + \gamma c(s,\hat{s})].
\end{aligned}
\tag{14}
$$

Finally, substituting Equation 14 back into Equation 13, we obtain the RHS of the lemma as

$$
\text{LHS} = \mathbb{E}_{s \sim P} \left[ \inf_{\hat{s} \in \mathcal{S}} (f(\hat{s}) + \gamma c(s,\hat{s})) \right] = \text{RHS}.
$$

This concludes the proof. □

Now we are ready to give our formal proof for Proposition 4.1. We restate it as follows.

**Proposition B.1** (Proposition 4.1). *Let $\mathcal{M}_\xi$ be the Wasserstein dynamics uncertainty set. Then we have*

$$
\min_{\widehat{T} \in \mathcal{M}_\xi} \mathbb{E}_{s' \sim \widehat{T}(\cdot|s,a)} \widehat{V}(s') = \mathbb{E}_{s' \sim \widehat{T}_{\text{MLE}}(\cdot|s,a)} \left[ \min_{\hat{s} \in U_\xi(s')} \widehat{V}(\hat{s}) \right].
\tag{15}
$$

*Proof.* We start with the LHS and prove its equivalence to the RHS.

The LHS is a constrained optimization problem that can be solved using the Lagrange multiplier method. We define the Lagrange function as

$$
\mathcal{L}(\widehat{T}, \lambda) = \mathbb{E}_{\hat{s}' \sim \widehat{T}(\cdot|s,a)} \widehat{V}(\hat{s}') + \lambda \left( \mathcal{W}(\widehat{T}, \widehat{T}_{\text{MLE}}) - \xi \right).
$$

The LHS is equivalent to solving the dual problem as

$$\text{LHS} = \sup_{\lambda \geq 0} \inf_{\widehat{T}} \mathcal{L}(\widehat{T}, \lambda). \tag{16}$$

For a fixed $\lambda$, we solve the inner minimization problem $\inf_{\widehat{T}} \mathcal{L}(\widehat{T}, \lambda)$ as

$$\inf_{\widehat{T}} \mathcal{L}(\widehat{T}, \lambda) = \inf_{\widehat{T}} \left( \mathbb{E}_{\hat{s}' \sim \widehat{T}(\cdot|s,a)} \widehat{V}(\hat{s}') + \lambda \mathcal{W}(\widehat{T}, \widehat{T}_{\text{MLE}}) \right) - \lambda \xi. \tag{17}$$

According to Lemma B.1, we have

$$\inf_{\widehat{T}} \left( \mathbb{E}_{\hat{s}' \sim \widehat{T}(\cdot|s,a)} \widehat{V}(\hat{s}') + \lambda \mathcal{W}(\widehat{T}, \widehat{T}_{\text{MLE}}) \right) = \mathbb{E}_{s' \sim \widehat{T}_{\text{MLE}}(\cdot|s,a)} \left[ \inf_{\hat{s}'} \left( \widehat{V}(\hat{s}') + \lambda c(s', \hat{s}') \right) \right]. \tag{18}$$

Substituting Equation 18 into Equation 17 and Equation 16, we obtain the dual reformulation of LHS as

$$\text{LHS} = \sup_{\lambda \geq 0} \left\{ \mathbb{E}_{s' \sim \widehat{T}_{\text{MLE}}(\cdot|s,a)} \left[ \inf_{\hat{s}'} \left( \widehat{V}(\hat{s}') + \lambda c(s', \hat{s}') \right) \right] - \lambda \xi \right\}. \tag{19}$$

The RHS in Equation 19 is exactly the Lagrange dual reformulation of the RHS in Equation 15. This implies Equation 15 holds, which concludes the proof. $\qquad\square$

## B.2 PROOF OF PROPOSITION 4.2

*Proof.* The standard Bellman operator is defined as

$$\mathcal{T}Q(s,a) = r + \gamma \mathbb{E}_{s' \sim \widehat{T}_{\text{MLE}}(\cdot|s,a), a' \sim \pi(\cdot|s')} Q(s', a').$$

For Bellman updates using $\mathcal{D}_{\text{model}}$, the Bellman operator could be expressed as

$$\mathcal{T}_{\text{model}} Q(s,a) = r + \gamma \mathbb{E}_{s' \sim \widehat{T}_\psi(\cdot|s,a), a' \sim \pi(\cdot|s')} Q(s', a').$$

For any $(s,a) \in \mathcal{D}_{\text{model}}$, we have

$$\mathbb{E}_{s' \sim \widehat{T}_\psi(\cdot|s,a), a' \sim \pi(\cdot|s')} Q(s', a') \leq \mathbb{E}_{s' \sim \widehat{T}_{\text{MLE}}(\cdot|s,a)} \left[ \min_{\tilde{s}' \in U_\xi(s'), a' \sim \pi(\cdot|\tilde{s}')} Q(\tilde{s}', a') \right] + \epsilon_1$$
$$\leq \mathbb{E}_{s' \sim \widehat{T}_{\text{MLE}}(\cdot|s,a), a' \sim \pi(\cdot|s')} Q(s', a') + \epsilon_1.$$

Therefore, we have

$$\mathcal{T}_{\text{model}} Q(s,a) \leq \mathcal{T}Q(s,a) + \gamma \epsilon_1, \qquad \forall (s,a) \in \mathcal{D}_{\text{model}}.$$

Let $\widehat{Q}^k$ denote the Q-value at iteration $k$ by applying $\mathcal{T}_{\text{model}}$, and $Q^k$ denote the $k$-iteration Q-value by applying $\mathcal{T}$. $Q^0$ represents the initial Q-value. After one iteration, we have

$$\widehat{Q}^1(s,a) \leq Q^1(s,a) + \frac{1-\gamma}{1-\gamma} \cdot \gamma \epsilon_1.$$

Suppose at iteration $k$, we have

$$\widehat{Q}^k(s,a) \leq Q^k(s,a) + \frac{1-\gamma^k}{1-\gamma} \gamma \epsilon_1, \qquad \forall k \in \mathbb{Z}^+. \tag{20}$$

Then for iteration $k+1$, we have

$$\widehat{Q}^{k+1}(s,a) = \mathcal{T}_{\text{model}} \widehat{Q}^k(s,a) \leq \mathcal{T} \widehat{Q}^k(s,a) + \gamma \epsilon_1.$$

In the meantime, we have

$$\mathcal{T} \widehat{Q}^k(s,a) \leq \mathcal{T} \left( Q^k(s,a) + \frac{1-\gamma^k}{1-\gamma} \gamma \epsilon_1 \right)$$
$$= r + \gamma \mathbb{E}_{s' \sim \widehat{T}_{\text{MLE}}(\cdot|s,a)} \left[ Q^k(s,a) + \frac{1-\gamma^k}{1-\gamma} \gamma \epsilon_1 \right]$$
$$= Q^{k+1} + \frac{1-\gamma^k}{1-\gamma} \gamma^2 \epsilon_1.$$

Therefore, we have

$$\widehat{Q}^{k+1}(s,a) \leq Q^{k+1} + \frac{1-\gamma^{k+1}}{1-\gamma}\gamma\epsilon_1.$$

Therefore, Equation 20 also holds for iteration $k+1$, and thus holds for all $k \in \mathbb{Z}^+$. If we set $k \to \infty$, then $\widehat{Q}$ and $Q$ would converge to the fixed points, and we have

$$\widehat{Q}(s,a) \leq Q_{\text{true}}(s,a) + \frac{\gamma\epsilon_1}{1-\gamma} \tag{21}$$

In the meantime, we also have

$$\mathbb{E}_{s'\sim\widehat{T}_\psi(\cdot|s,a),a'\sim\pi(\cdot|s')}Q(s',a') \geq \mathbb{E}_{s'\sim\widehat{T}_{\text{MLE}}(\cdot|s,a)}\left[\min_{\tilde{s}'\in U_\xi(s'),a'\sim\pi(\cdot|\tilde{s}')}Q(\tilde{s}',a')\right] - \epsilon_1$$
$$\geq \mathbb{E}_{s'\sim\widehat{T}_{\text{MLE}}(\cdot|s,a),a'\sim\pi(\cdot|s')}Q(s',a') - (\epsilon_1 + \epsilon_2).$$

Following a similar derivation process as above, we obtain

$$\widehat{Q}(s,a) \geq Q_{\text{true}}(s,a) - \frac{\gamma(\epsilon_1 + \epsilon_2)}{1-\gamma}. \tag{22}$$

Combining the results of Equation 21 and Equation 22, we conclude the proof. $\qquad\square$

### B.3 PROOF OF COROLLARY 4.1

*Proof.* The proof is straightforward.

$$\epsilon_1 = \max_{(s,a)\in\mathcal{D}_{\text{model}}} \left|\mathbb{E}_{\hat{s}'\sim\widehat{T}_\psi(\cdot|s,a),s'\sim\widehat{T}_{\text{MLE}}(\cdot|s,a)}\left[\widehat{V}(\hat{s}') - \min_{\tilde{s}'\in U_\xi(s')}\widehat{V}(\tilde{s}')\right]\right|$$

$$= \max_{(s,a)\in\mathcal{D}_{\text{model}}} \left|\mathbb{E}_{\hat{s}'\sim\widehat{T}_\psi(\cdot|s,a),s'\sim\widehat{T}_{\text{MLE}}(\cdot|s,a)}\left[\widehat{V}(\hat{s}') - \widehat{V}(s') + \widehat{V}(s') - \min_{\tilde{s}'\in U_\xi(s')}\widehat{V}(\tilde{s}')\right]\right|$$

$$\leq \max_{(s,a)\in\mathcal{D}_{\text{model}}} \mathbb{E}_{\hat{s}'\sim\widehat{T}_\psi(\cdot|s,a),s'\sim\widehat{T}_{\text{MLE}}(\cdot|s,a)}\left[\left|\widehat{V}(\hat{s}') - \widehat{V}(s')\right| + \left|\widehat{V}(s') - \min_{\tilde{s}'\in U_\xi(s')}\widehat{V}(\tilde{s}')\right|\right]$$

$$\leq L_V \cdot (\epsilon + \xi).$$

This concludes the proof. $\qquad\square$

### B.4 PROOF OF PROPOSITION 4.3

**Definition B.1** (Lipschitz Smoothness). *A function $f(x) : \mathbb{R}^n \to \mathbb{R}$ is L-Lipschitz smooth if the following inequality holds*

$$\|\nabla f(x_1) - \nabla f(x_2)\| \leq L\|x_1 - x_2\|, \forall x_1, x_2 \in \mathbb{R}^n. \tag{23}$$

We then have the following Lemmas:

**Lemma B.2** (Nesterov et al. (2018)). *If function $f(x) : \mathbb{R}^n \to \mathbb{R}$ is L-Lipschitz smooth, then we have*

$$\left|f(x_2) - f(x_1) - \nabla f(x_1)^\mathsf{T}(x_2 - x_1)\right| \leq \frac{L}{2}\|x_2 - x_1\|^2, \forall x_1, x_2 \in \mathbb{R}^n. \tag{24}$$

**Lemma B.3.** *Assume the outer loss $\mathcal{L}_{\text{RVL}}$ is L-Lipschitz smooth, and the gradient of $\mathcal{L}_{\text{RVL}}$, $\mathcal{L}_{\text{SL}}$ are bounded by $\rho$. Let $w_\nu$ be twice differentiable, with its gradient and Hessian bounded by $\delta$ and $\mathcal{B}$, respectively. Then $\mathcal{L}_{\text{RVL}}$ is $\rho^2(L\delta^2 + \mathcal{B})$-Lipschitz smooth w.r.t. $\nu$.*

*Proof.* Recall that the gradient of the outer loss $\mathcal{L}_{\text{RVL}}(\psi(\nu))$ w.r.t. $\nu$ is

$$\nabla_\nu\mathcal{L}_{\text{RVL}}(\psi^{(t+1)}(\nu^{(t)})) = \nabla_\psi\mathcal{L}_{\text{RVL}}(\psi^{(t+1)}(\nu^{(t)})) \cdot \nabla_\nu\psi^{(t+1)}(\nu^{(t)})$$
$$= h \cdot \nabla_\nu w_{\nu^{(t)}}(s,a,s'), \tag{25}$$

where $h$ is defined in Equation 10. Taking the gradient of $\nu$ in both sides of Equation 25, we have

$$\nabla_\nu^2\mathcal{L}_{\text{RVL}}(\psi^{(t+1)}(\nu^{(t)})) = \underbrace{\nabla_\nu h \cdot \nabla_{\nu^{(t)}} w_\nu(s,a,s')}_{I_1} + \underbrace{h \cdot \nabla_\nu^2 w_{\nu^{(t)}}(s,a,s')}_{I_2}.$$

The norm of the first term $I_1$ could be bounded as follows:

$$
\begin{aligned}
&\|I_1\| \\
&= \|\nabla_\nu h \cdot \nabla_\nu w_{\nu^{(t)}}(s, a, s')\| \\
&\leq \beta_{1(t)}\delta \left\| \nabla_\psi \left( \nabla_\nu \mathcal{L}_{\mathrm{RVL}}(\psi^{(t+1)}(\nu^{(t)})) \right)^{\mathsf{T}} \nabla_\psi \log \left( \widehat{T}_{\psi^{(t)}}(s'|s, a) \right) \right\| \\
&= \beta_{1(t)}^2 \delta \left\| \nabla_\psi \left( \nabla_\psi \mathcal{L}_{\mathrm{RVL}}(\psi^{(t+1)}) \nabla_\psi \log \left( \widehat{T}_{\psi^{(t)}}(s'|s, a) \right) \nabla_\nu w_{\nu^{(t)}}(s, a, s') \right)^{\mathsf{T}} \nabla_\psi \log \left( \widehat{T}_{\psi^{(t)}}(s'|s, a) \right) \right\| \\
&= \beta_{1(t)}^2 \delta \left\| \left( \nabla_\psi^2 \mathcal{L}_{\mathrm{RVL}}(\psi^{(t+1)}) \nabla_\psi \log \left( \widehat{T}_{\psi^{(t)}}(s'|s, a) \right) \nabla_\nu w_{\nu^{(t)}}(s, a, s') \right)^{\mathsf{T}} \nabla_\psi \log \left( \widehat{T}_{\psi^{(t)}}(s'|s, a) \right) \right\| \\
&\leq \beta_{1(t)}^2 L \rho^2 \delta^2.
\end{aligned}
$$

The norm of the second term $I_2$ could be bounded as below:

$$
\begin{aligned}
\|I_2\| &= \left\| h \cdot \nabla_\nu^2 w_{\nu^{(t)}}(s, a, s') \right\| \\
&= \beta_{1(t)} \left\| \nabla_\psi \mathcal{L}_{\mathrm{RVL}}(\psi^{(t+1)})^{\mathsf{T}} \nabla_\psi \log \left( \widehat{T}_{\psi^{(t)}}(s'|s, a) \right) \nabla_\nu^2 w_{\nu^{(t)}}(s, a, s') \right\| \\
&\leq \beta_{1(t)} \mathcal{B} \rho^2.
\end{aligned}
$$

Combining the above results of $\|I_1\|$ and $\|I_2\|$, we could bound the norm of $\nabla_\nu^2 \mathcal{L}_{\mathrm{RVL}}(\psi^{(t+1)}(\nu^{(t)}))$ as follows:

$$
\begin{aligned}
\left\| \nabla_\nu^2 \mathcal{L}_{\mathrm{RVL}}(\psi^{(t+1)}(\nu^{(t)})) \right\| &= \|I_1 + I_2\| \\
&\leq \|I_1\| + \|I_2\| \\
&\leq \beta_{1(t)} \rho^2 (\beta_{1(t)} L \delta^2 + \mathcal{B}) \\
&\leq \rho^2 (L \delta^2 + \mathcal{B}).
\end{aligned}
$$

According to Lagrange mean value theorem, for $\forall \nu_1, \nu_2$, we have

$$
\|\nabla_\nu \mathcal{L}_{\mathrm{RVL}}(\psi(\nu_1)) - \nabla_\nu \mathcal{L}_{\mathrm{RVL}}(\psi(\nu_2))\| \leq L' \|\nu_1 - \nu_2\|,
$$

where $L' = \rho^2 (L \delta^2 + \mathcal{B})$. This concludes the proof. $\qquad\square$

Then we prove our proposition 4.3. We restate it as follows.

**Proposition B.2** (Proposition 4.3). *Assume $\mathcal{L}_{\mathrm{RVL}}$ and $\mathcal{L}_{\mathrm{WSL}}$ are L-Lipschitz smooth, and the gradient of $\mathcal{L}_{\mathrm{RVL}}$, $\mathcal{L}_{\mathrm{SL}}$ and $\mathcal{L}_{\mathrm{WSL}}$ are bounded by $\rho$. Let $w_\nu$ be twice differentiable, with its gradient and Hessian bounded by $\delta$ and $\mathcal{B}$, respectively. Denote $L' = \rho^2 (L \delta^2 + \mathcal{B})$, and the total steps as $K$. For some $c_1, c_2 > 0$, we assume the learning rate of the inner and outer loop $\beta_1$ and $\beta_2$ satisfies: $\frac{c_2}{\sqrt{K}} \leq \beta_1, \beta_2 \leq \min\{\frac{c_1}{K}, \frac{1}{L}, \frac{1}{L'}\}$, where $\frac{c_2}{\sqrt{K}} \leq \min\{\frac{c_1}{K}, \frac{1}{L}, \frac{1}{L'}\}, \frac{c_2}{\sqrt{K}} < 1, \frac{c_1}{K} < 1$. Then the outer loss $\mathcal{L}_{\mathrm{RVL}}$ and inner loss $\mathcal{L}_{\mathrm{WSL}}$ can achieve a convergence rate of $\mathcal{O}\left(\frac{1}{\sqrt{K}}\right)$:*

$$
\min_{0 \leq t \leq K-1} \mathbb{E}\left[ \left\| \nabla_\nu \mathcal{L}_{\mathrm{RVL}}(\psi^{(t+1)}(\nu^{(t)})) \right\|^2 \right] \leq \mathcal{O}\left(\frac{1}{\sqrt{K}}\right), \tag{26}
$$

$$
\min_{0 \leq t \leq K-1} \mathbb{E}\left[ \left\| \nabla_\psi \mathcal{L}_{\mathrm{WSL}}(\psi^{(t)}(\nu^{(t)})) \right\|^2 \right] \leq \mathcal{O}\left(\frac{1}{\sqrt{K}}\right). \tag{27}
$$

*Proof.* We first prove Equation 26. Consider the difference in the outer loss between the $t$-th and the $t+1$-th iteration:

$$
\begin{aligned}
&\mathcal{L}_{\mathrm{RVL}}(\psi^{(t+2)}(\nu^{(t+1)})) - \mathcal{L}_{\mathrm{RVL}}(\psi^{(t+1)}(\nu^{(t)})) \\
&= \underbrace{\mathcal{L}_{\mathrm{RVL}}(\psi^{(t+2)}(\nu^{(t+1)})) - \mathcal{L}_{\mathrm{RVL}}(\psi^{(t+1)}(\nu^{(t+1)}))}_{I_1} + \underbrace{\mathcal{L}_{\mathrm{RVL}}(\psi^{(t+1)}(\nu^{(t+1)})) - \mathcal{L}_{\mathrm{RVL}}(\psi^{(t+1)}(\nu^{(t)}))}_{I_2}.
\end{aligned}
$$

According to Lemma B.2 and the assumption that $\mathcal{L}_{\mathrm{RVL}}$ is Lipschitz smooth, we could bound the first term $I_1$ by

$$
I_1 \leq \underbrace{\nabla_\psi \mathcal{L}_{\mathrm{RVL}}(\psi^{(t+1)}(\nu^{(t+1)}))^{\mathsf{T}}(\psi^{(t+2)}(\nu^{(t+1)}) - \psi^{(t+1)}(\nu^{t+1}))}_{I_3}
$$

$$
+ \underbrace{\frac{L}{2}\left\|\psi^{(t+2)}(\nu^{(t+1)}) - \psi^{(t+1)}(\nu^{(t+1)})\right\|^2}_{I_4}.
$$

For term $I_3$, since $\psi^{(t+2)}(\nu^{(t+1)})$ and $\psi^{(t+1)}(\nu^{(t+1)})$ have the relation of

$$
\psi^{(t+2)}(\nu^{(t+1)}) = \psi^{(t+1)}(\nu^{(t+1)}) - \beta_{1(t+1)}\nabla_\psi \mathcal{L}_{\mathrm{WSL}}(\psi^{(t+1)}),
$$

we could bound $I_3$ by

$$
I_3 \leq \left\|\nabla_\psi \mathcal{L}_{\mathrm{RVL}}(\psi^{(t+1)}(\nu^{(t+1)}))\right\| \cdot \left\|\psi^{(t+2)}(\nu^{(t+1)}) - \psi^{(t+1)}(\nu^{(t+1)})\right\|
$$

$$
\leq \rho \cdot \left\|\beta_{1(t+1)}\nabla_\psi \mathcal{L}_{\mathrm{WSL}}(\psi^{(t+1)})\right\|
$$

$$
\leq \beta_{1(t+1)}\rho^2.
$$

Similarly, $I_4$ is bounded by

$$
I_4 = \frac{L}{2}\left\|\beta_{1(t+1)}\nabla_\psi \mathcal{L}_{\mathrm{WSL}}(\psi^{(t+1)})\right\|^2
$$

$$
\leq \frac{L}{2}\beta_{1(t+1)}^2\rho^2.
$$

Therefore, we have

$$
I_1 \leq I_3 + I_4 \leq \beta_{1(t+1)}\rho^2 + \frac{L}{2}\beta_{1(t+1)}^2\rho^2. \tag{28}
$$

For term $I_2$, we regard $\mathcal{L}_{\mathrm{RVL}}(\psi(\nu))$ as a function w.r.t. $\nu$. By using Lemma B.2 and Lemma B.3, we have

$$
I_2 \leq \nabla_\nu \mathcal{L}_{\mathrm{RVL}}(\psi^{(t+1)}(\nu^{(t)}))^{\mathsf{T}}(\nu^{(t+1)} - \nu^{(t)}) + \frac{L'}{2}\left\|\nu^{(t+1)} - \nu^{(t)}\right\|^2
$$

$$
= -\beta_{2(t)}\nabla_\nu \mathcal{L}_{\mathrm{RVL}}(\psi^{(t+1)}(\nu^{(t)}))^{\mathsf{T}}\nabla_\nu \mathcal{L}_{\mathrm{RVL}}(\psi^{(t+1)}(\nu^{(t)})) + \frac{L'\beta_{2(t)}^2}{2}\left\|\nabla_\nu \mathcal{L}_{\mathrm{RVL}}(\psi^{(t+1)}(\nu^{(t)}))\right\|^2
$$

$$
= -\left(\beta_{2(t)} - \frac{L'\beta_{2(t)}^2}{2}\right)\left\|\nabla_\nu \mathcal{L}_{\mathrm{RVL}}(\psi^{(t+1)}(\nu^{(t)}))\right\|^2.
$$

$$
\tag{29}
$$

The first equality holds due to

$$
\nu^{(t+1)} = \nu^{(t)} - \beta_{2(t)} \cdot \nabla_\nu \mathcal{L}_{\mathrm{RVL}}(\psi^{(t+1)}(\nu^{(t)})).
$$

Combining the results of Equation 28 and Equation 29, we have

$$
\mathcal{L}_{\mathrm{RVL}}(\psi^{(t+2)}(\nu^{(t+1)})) - \mathcal{L}_{\mathrm{RVL}}(\psi^{(t+1)}(\nu^{(t)}))
$$

$$
\leq \beta_{1(t+1)}\rho^2 + \frac{L}{2}\beta_{1(t+1)}^2\rho^2 - \left(\beta_{2(t)} - \frac{L'\beta_{2(t)}^2}{2}\right)\left\|\nabla_\nu \mathcal{L}_{\mathrm{RVL}}(\psi^{(t+1)}(\nu^{(t)}))\right\|^2.
$$

Adjusting the order of the inequality and summing both sides from $t = 0$ to $K - 1$, we have

$$
\sum_{t=0}^{K-1}\left(\beta_{2(t)} - \frac{L'\beta_{2(t)}^2}{2}\right)\left\|\nabla_\nu \mathcal{L}_{\mathrm{RVL}}(\psi^{(t+1)}(\nu^{(t)}))\right\|^2
$$

$$
\leq \mathcal{L}_{\mathrm{RVL}}(\psi^{(1)}(\nu^{(0)})) - \mathcal{L}_{\mathrm{RVL}}(\psi^{(K+1)}(\nu^{(K)})) + \sum_{t=0}^{K-1}\left(\beta_{1(t+1)}\rho^2 + \frac{L\beta_{1(t+1)}^2\rho^2}{2}\right)
$$

$$
\leq \mathcal{L}_{\mathrm{RVL}}(\psi^{(1)}(\nu^{(0)})) + \sum_{t=0}^{K-1}\left(\beta_{1(t+1)}\rho^2 + \frac{L\beta_{1(t+1)}^2\rho^2}{2}\right).
$$

Therefore, we have

$$
\min_{0 \le t \le K-1} \mathbb{E}\left[\left\|\nabla_\nu \mathcal{L}_{\mathrm{RVL}}(\psi^{(t+1)}(\nu^{(t)}))\right\|\right]
$$

$$
\le \frac{\sum_{t=0}^{K-1}(\beta_{2(t)} - \frac{L'\beta_{2(t)}^2}{2})\left\|\nabla_\nu \mathcal{L}_{\mathrm{RVL}}(\psi^{(t+1)}(\nu^{(t)}))\right\|^2}{\sum_{t=0}^{K-1}(\beta_{2(t)} - \frac{L'\beta_{2(t)}^2}{2})}
$$

$$
\le \frac{2\mathcal{L}_{\mathrm{RVL}}(\psi^{(1)}(\nu^{(0)})) + \sum_{t=0}^{K-1}(2\beta_{1(t+1)}\rho^2 + L\beta_{1(t+1)}^2\rho^2)}{\sum_{t=0}^{K-1}(2\beta_{2(t)} - L'\beta_{2(t)}^2)}
$$

$$
\le \frac{1}{\sum_{t=0}^{K-1}\beta_{2(t)}}\left[2\mathcal{L}_{\mathrm{RVL}}(\psi^{(1)}(\nu^{(0)})) + \sum_{t=0}^{K-1}\beta_{1(t+1)}\rho^2(2 + L\beta_{1(t+1)})\right]
$$

$$
\le \frac{1}{K\min_{0\le t\le K-1}\beta_{2(t)}}\left[2\mathcal{L}_{\mathrm{RVL}}(\psi^{(1)}(\nu^{(0)})) + K\max_{0\le t\le K-1}\beta_{1(t+1)}\rho^2(2 + L)\right]
$$

$$
= \frac{2\mathcal{L}_{\mathrm{RVL}}(\psi^{(1)}(\nu^{(0)}))}{K\min_{0\le t\le K-1}\beta_{2(t)}} + \frac{\max_{0\le t\le K-1}\beta_{1(t+1)}\rho^2(2 + L)}{\min_{0\le t\le K-1}\beta_{2(t)}}
$$

$$
\le \frac{2\mathcal{L}_{\mathrm{RVL}}(\psi^{(1)}(\nu^{(0)}))}{c_2\sqrt{K}} + \frac{c_1\rho^2(2 + L)}{c_2\sqrt{K}}
$$

$$
= \mathcal{O}\left(\frac{1}{\sqrt{K}}\right).
$$

The third inequality holds from $\sum_{t=0}^{K-1}\beta_{2(t)} \le \sum_{t=0}^{K-1}(2\beta_{2(t)} - L'\beta_{2(t)}^2)$ given $\beta_{2(t)} \le \frac{1}{L'}$.

The proof of Equation 27 is similar to Equation 26. For completeness, we also present the proof procedure. Consider the difference in the inner loss between the $t$-th and $t + 1$-th iterations:

$$
\mathcal{L}_{\mathrm{WSL}}(\psi^{(t+1)}, \nu^{(t+1)}) - \mathcal{L}_{\mathrm{WSL}}(\psi^{(t)}, \nu^{(t)})
$$
$$
= \underbrace{\mathcal{L}_{\mathrm{WSL}}(\psi^{(t+1)}, \nu^{(t+1)}) - \mathcal{L}_{\mathrm{WSL}}(\psi^{(t+1)}, \nu^{(t)})}_{I_1} + \underbrace{\mathcal{L}_{\mathrm{WSL}}(\psi^{(t+1)}, \nu^{(t)}) - \mathcal{L}_{\mathrm{WSL}}(\psi^{(t)}, \nu^{(t)})}_{I_2}.
$$

For term $I_1$, according to Lemma B.2, we have

$$
I_1 \le \nabla_\nu \mathcal{L}_{\mathrm{WSL}}(\psi^{(t+1)}, \nu^{(t)})^\mathsf{T}(\nu^{(t+1)} - \nu^{(t)}) + \frac{L}{2}\left\|\nu^{(t+1)} - \nu^{(t)}\right\|^2
$$

$$
= -\beta_{2(t)}\nabla_\nu \mathcal{L}_{\mathrm{WSL}}(\psi^{(t+1)}, \nu^{(t)})^\mathsf{T}\nabla_\nu \mathcal{L}_{\mathrm{RVL}}(\psi^{(t+1)}(\nu^{(t)})) + \frac{L\beta_{2(t)}^2}{2}\left\|\nabla_\nu \mathcal{L}_{\mathrm{RVL}}(\psi^{(t+1)}(\nu^{(t)}))\right\|^2
$$

$$
\le \beta_{2(t)}\rho^2 + \frac{L}{2}\beta_{2(t)}^2\rho^2.
$$

For term $I_2$, we have

$$
I_2 \le \nabla_\psi \mathcal{L}_{\mathrm{WSL}}(\psi^{(t)}, \nu^{(t)})^\mathsf{T}(\psi^{(t+1)} - \psi^{(t)}) + \frac{L}{2}\left\|\psi^{(t+1)} - \psi^{(t)}\right\|^2
$$

$$
= -\beta_{1(t)}\nabla_\psi \mathcal{L}_{\mathrm{WSL}}(\psi^{(t)}, \nu^{(t)})^\mathsf{T}\nabla_\psi \mathcal{L}_{\mathrm{WSL}}(\psi^{(t)}, \nu^{(t)}) + \frac{L\beta_{1(t)}^2}{2}\left\|\nabla_\psi \mathcal{L}_{\mathrm{SL}}(\psi^{(t)}, \nu^{(t)})\right\|^2
$$

$$
= -(\beta_{1(t)} - \frac{L\beta_{1(t)}^2}{2})\left\|\nabla_\psi \mathcal{L}_{\mathrm{WSL}}(\psi^{(t)}, \nu^{(t)})\right\|^2.
$$

Therefore, we have

$$
\mathcal{L}_{\mathrm{WSL}}(\psi^{(t+1)}, \nu^{(t+1)}) - \mathcal{L}_{\mathrm{WSL}}(\psi^{(t)}, \nu^{(t)})
$$
$$
\le \beta_{2(t)}\rho^2 + \frac{L}{2}\beta_{2(t)}^2\rho^2 - \left(\beta_{1(t)} - \frac{L\beta_{1(t)}^2}{2}\right)\left\|\nabla_\psi \mathcal{L}_{\mathrm{WSL}}(\psi^{(t)}, \nu^{(t)})\right\|^2.
$$

Rearranging the inequality and summing both sides from $t = 0$ to $K - 1$, we have

$$\sum_{t=0}^{K-1} \left( \beta_{1(t)} - \frac{L\beta_{1(t)}^2}{2} \right) \left\| \nabla_\psi \mathcal{L}_{\text{WSL}}(\psi^{(t)}, \nu^{(t)}) \right\|^2$$

$$\leq \mathcal{L}_{\text{WSL}}(\psi^{(0)}, \nu^{(0)}) - \mathcal{L}_{\text{WSL}}(\psi^{(K)}, \nu^{(K)}) + \sum_{t=0}^{K-1} \beta_{2(t)} \rho^2 + \frac{L}{2} \beta_{2(t)}^2 \rho^2$$

$$\leq \mathcal{L}_{\text{WSL}}(\psi^{(0)}, \nu^{(0)}) + \sum_{t=0}^{K-1} \beta_{2(t)} \rho^2 + \frac{L}{2} \beta_{2(t)}^2 \rho^2.$$

Therefore, we have

$$\min_{0 \leq t \leq T-1} \mathbb{E} \left[ \left\| \nabla_\psi \mathcal{L}_{\text{WSL}}(\psi^{(t)}, \nu^{(t)}) \right\|^2 \right]$$

$$\leq \frac{\sum_{t=0}^{K-1} \left( \beta_{1(t)} - \frac{L\beta_{1(t)}^2}{2} \right) \left\| \nabla_\psi \mathcal{L}_{\text{WSL}}(\psi^{(t)}, \nu^{(t)}) \right\|^2}{\sum_{t=0}^{K-1} \left( \beta_{1(t)} - \frac{L\beta_{1(t)}^2}{2} \right)}$$

$$\leq \frac{2\mathcal{L}_{\text{WSL}}(\psi^{(0)}, \nu^{(0)}) + \sum_{t=0}^{K-1} 2\beta_{2(t)} \rho^2 + L\beta_{2(t)}^2 \rho^2}{\sum_{t=0}^{K-1} \left( 2\beta_{1(t)} - L\beta_{1(t)}^2 \right)}$$

$$\leq \frac{2\mathcal{L}_{\text{WSL}}(\psi^{(0)}, \alpha^{(0)}) + \sum_{t=0}^{K-1} 2\beta_{2(t)} \rho^2 + L\beta_{2(t)}^2 \rho^2}{\sum_{t=0}^{K-1} \beta_{1(t)}}$$

$$\leq \frac{1}{K \min_{0 \leq t \leq K-1} \beta_{1(t)}} \left[ 2\mathcal{L}_{\text{WSL}}(\psi^{(0)}, \nu^{(0)}) + K \max_{0 \leq t \leq K-1} \beta_{2(t)} \rho^2 (2 + L) \right]$$

$$\leq \frac{2\mathcal{L}_{\text{WSL}}(\psi^{(0)}, \nu^{(0)})}{c_2 \sqrt{K}} + \frac{c_1 \rho^2 (2 + L)}{c_2 \sqrt{K}}$$

$$= \mathcal{O}(\frac{1}{\sqrt{K}}),$$

where the third inequality holds since $\sum_{t=0}^{K-1} \beta_{1(t)} \leq \sum_{t=0}^{K-1} (2\beta_{1(t)} - L\beta_{1(t)}^2)$ given $\beta_{1(t)} \leq \frac{1}{L}$. This concludes the proof. $\square$

## C  BENCHMARK SETTINGS

In this part, we provide details of the benchmarks and datasets we use for evaluating our method in our paper. The offline datasets are taken from the D4RL (Fu et al., 2020) and NeoRL (Qin et al., 2022) benchmarks, two popular benchmarks designed for evaluating offline RL algorithms.

### C.1  D4RL

For the D4RL benchmark, we mainly evaluate our method on two kinds of datasets: MuJoCo datasets and Antmaze datasets. We first introduce the two datasets.

MuJoCo datasets are collected through interactions with continuous control tasks in Gym (Brockman et al., 2016) simulated by MuJoCo (Todorov et al., 2012). The tasks we use are `halfcheetah`, `hopper` and `walker2d`, as illustrated in Figure 4. For each task, we use the four types of datasets: (1) **Random**: data collected with a random policy. (2) **Medium**: 1M samples collected by an early-stopped SAC policy. (3) **Medium-Replay**: 1M samples from the replay buffer of the agent trained up to the performance of a medium-level agent. (4) **Medium-Expert**: 50-50 split of medium-level data and expert-level data. The MuJoCo dataset version we use in our work is "-v2". The metric we use for evaluating the agent's performance on MuJoCo

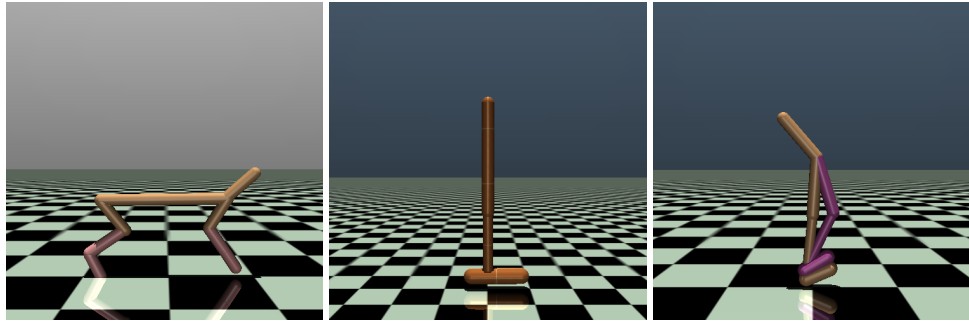

Figure 4: D4RL MuJoCo tasks. **Left:** halfcheetah, **Middle:** hopper, **Right:** walker2d.

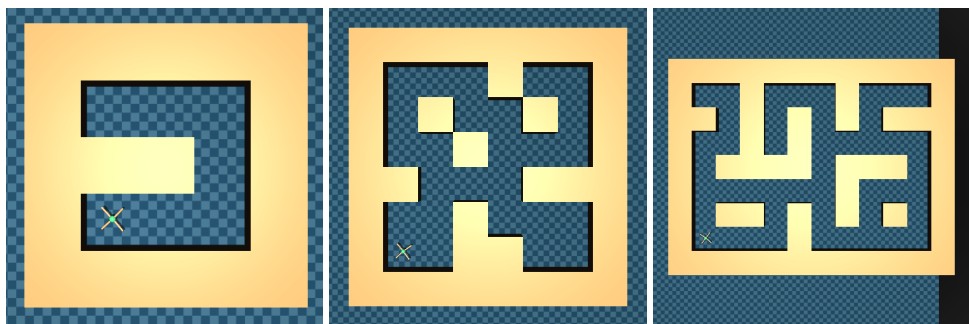

Figure 5: D4RL Antmaze tasks. **Left:** umaze, **Middle:** medium, **Right:** large.

tasks is Normalized Score (NS). NS is computed as follows.

$$\text{NS} = \frac{J_\pi - J_{\text{random}}}{J_{\text{expert}} - J_{\text{random}}} \times 100\% \tag{30}$$

where $J_\pi$ is the performance of the evaluated policy, $J_{\text{random}}$ is the performance of the random policy, $J_{\text{expert}}$ is the performance of the expert policy.

Antmaze datasets are collected in Antmaze tasks. In Antmaze tasks, an 8-DOF "Ant" quadraped robot is required to reach a goal location. Antmaze tasks are more challenging than MuJoCo tasks for offline RL algorithms due to their sparse reward setting. There are three maze layouts contained in Antmaze tasks: `umaze`, `medium`, `large`, as shown in Figure 5. The datasets are collected in three flavors: *(i)* the robot needs to reach a specified goal from a fixed start point (`antmaze-umaze-v0`). *(ii)* the robot is required to reach a random goal from a random start point (the `diverse` datasets). *(iii)* the robot is commanded to reach specific locations from a different set of specific start locations (the `play` datasets). In our work, we use the six Antmaze datasets: `antmaze-umaze`, `antmaze-umaze-diverse`, `antmaze-medium-diverse`, `antmaze-medium-play`, `antmaze-large-diverse`, `antmaze-large-play`. The dataset version we use is "-v0". The metric we use for evaluating the policy performance on Antmaze tasks is the success rate of task completion.

## C.2 NEORL

NeoRL is a benchmark that aims to simulate real-world scenarios by collecting datasets with a more conservative policy, which more aligns with the real-world data-collection strategy. NeoRL datasets contain narrow and limited data, making it challenging for offline RL methods. There are several tasks on the NeoRL benchmark. Following previous studies (Sun et al., 2023; Lin et al.), we focus on three tasks (`HalfCheetah-v3`, `Hopper-v3`, `Walker2d-v3`). For each task, we use three types of datasets collected with policies of different qualities (`low`, `medium`, `high`). NeoRL provides different numbers of trajectories (100, 1000, 10000) as training data for each task. We select 1000 trajectories for our experiments. The evaluation metric is the Normalized Score, the same metric used for MuJoCo tasks in D4RL.

# D IMPLEMENTATION DETAILS

In this part, we present the details of baseline implementations, ROMI implementations, and hyper-parameter setup.

## D.1 BASELINE IMPLEMENTATION

We elaborate on the baseline implementation details on D4RL and NeoRL benchmarks, respectively.

For D4RL datasets, our baselines include CQL (Kumar et al., 2020), IQL (Kostrikov et al., 2021), MOPO (Yu et al., 2020), RAMBO (Rigter et al., 2022), Count-MORL (Kim & Oh, 2023), MO-BILE (Sun et al., 2023). For Count-MORL, we retrain the algorithm using the official codebase[2] provided in the original paper. We note that there are three count estimation methods (`LC`, `AVG`, `UC`) chosen for Count-MORL, and we choose `UC` since the original paper reports that `UC` exhibits the best average performance. For other baselines including CQL, IQL, MOPO, RAMBO, and MO-BILE, we run the code in the codebase OfflineRL-Kit[3], which provides reliable implementations for different model-free and model-based offline RL algorithms.

For NeoRL datasets, our baselines include CQL, IQL, EDAC (An et al., 2021), MOPO, RAMBO, and MOBILE. Since the codebase OfflineRL-Kit has provided the implementations for these baselines, we directly use it as our baseline codebase.

For Antmaze tasks, we run RAMBO according to the hyperparameter setting in the original paper, since MOPO, COMBO, and MOBILE do not provide the detailed hyperparameter setting for Antmaze tasks, we directly copy the evaluation results of these algorithms as reported in (Lin et al., 2024).

It is worth noting that the default training steps in OfflineRL-Kit are 3M. To be consistent with the general training settings and ensure a fair comparison, we uniformly set the training steps for all algorithms to 1M.

## D.2 IMPLEMENTATION OF ROMI

We then provide the implementation details of ROMI from the following aspects:

**Codebase.** Since ROMI is an improved version of RAMBO, we choose OfflineRL-Kit as our codebase, and implement ROMI on top of the implementation of RAMBO.

**Dynamics Model Training.** Following previous works (Rigter et al., 2022; Sun et al., 2023; Kim & Oh, 2023), we represent the dynamics model as an ensemble of neural networks that output a Gaussian distribution over the next state given the current state and action:

$$\widehat{T}_\psi(s'|s,a) = \mathcal{N}(\mu_\psi(s,a), \Sigma_\psi(s,a)) \tag{31}$$

Note that we do not predict the reward $r$, since we assume the reward function $r$ is known as in (Sun et al., 2023), although $r$ can be considered as part of the model if unknown. Similar to RAMBO, we perform an initial maximum likelihood model training. The difference lies in that we fix the training epochs to 50, rather than stopping the training based on the held-out loss, since we find this could be quite time-consuming. After the initial training, ROMI updates all the ensemble members in a bi-level optimization framework. For each model rollout, we randomly select an ensemble member to simulate the rollout.

**Adaptive-weighting Network Training.** We represent the adaptive-weighting network as a simple Multi-Layer Perceptron (MLP) with 2 hidden layers. The input vector is a concatenation of $(s, a, s')$, and the output is a scalar representing the allocated weight for each training sample. To map the network's output to the interval $[a, b]$, we apply the $\tanh$ function and then perform an affine transformation (scaling and shifting) to regulate the final range:

$$\text{weight} = \tanh(\text{output}) \times \frac{b-a}{2} + \frac{a+b}{2} \tag{32}$$

---

[2]https://github.com/oh-lab/Count-MORL.git
[3]https://github.com/yihaosun1124/OfflineRL-Kit.git

In our experiment, we choose $a = 0.5$ and $b = 2$, that is, the assigned weight is constrained in the range $[0.5, 2]$. Since the performance of ROMI is already satisfactory, we do not further tune the values of $a$ and $b$.

**Policy Optimization.** The policy optimization method we adopt is SAC, and most hyperparameters follow its standard implementations. For each training step, we sample a batch of $256$ samples where $50\%$ of them are from the offline dataset $\mathcal{D}$, and another $50\%$ are from the model buffer $\mathcal{D}_{\mathrm{model}}$. We use automatic entropy tuning, where the entropy target is set to the standard heuristic $-\dim(\mathcal{A})$.

We present the base hyperparameter configurations for ROMI in Table 3 and the pseudo-code for ROMI in Algorithm 1. The key difference between ROMI and RAMBO lies in the training strategy for the dynamics model, as highlighted by the blue parts in the pseudo-code.

---

**Algorithm 1** ROMI

---

1: **Require:** Offline dataset $\mathcal{D}$, number of epochs $N$, maximum rollout length $H$, dynamics model pretraining steps $T_{\mathrm{model}}$, policy pretraining steps $T_{\mathrm{policy}}$, learning rate $\lambda$, real data ratio $f$

2: **Initialization:** model data buffer $\mathcal{D}_{\mathrm{model}} \leftarrow \varnothing$, ensemble dynamics model $\widehat{T}_\psi = \{\widehat{T}_{\psi_i}\}_{i=1}^M$, weighting network $w_\nu$, initial policy $\pi_\theta$, other components of SAC (Haarnoja et al., 2018) algorithm

3: **// Pre-train the dynamics model**

4: **for** step in 1 to $T_{\mathrm{model}}$ **do**

5: $\quad$ Compute loss: $\mathcal{L}_{\mathrm{SL}} = -\mathbb{E}_{(s,a,s') \in \mathcal{D}} \left[ \log \widehat{T}_\psi(s'|s,a) \right]$

6: $\quad$ Perform gradient descent: $\psi \leftarrow \psi - \lambda \nabla_\psi \mathcal{L}_{\mathrm{SL}}(\psi)$

7: **end for**

8: **// (Optional) Pre-train the policy**

9: **for** step in 1 to $T_{\mathrm{policy}}$ **do**

10: $\quad$ Compute loss: $\mathcal{L}_{\mathrm{BC}} = -\mathbb{E}_{(s,a) \in \mathcal{D}} \left[ \log \pi_\theta(s|a) \right]$

11: $\quad$ Perform gradient descent: $\theta \leftarrow \theta - \lambda \nabla_\theta \mathcal{L}_{\mathrm{BC}}(\theta)$

12: **end for**

13: **for** epoch from 1 to $N$ **do**

14: $\quad$ Sample an initial state $s_0$ from $\mathcal{D}$

15: $\quad$ **for** $h$ in 0 to $H-1$ **do**

16: $\quad\quad$ Take an action $a_h \sim \pi_\theta(\cdot|s_h)$

17: $\quad\quad$ Randomly pick an ensemble member $\widehat{T}$ from $\{\widehat{T}_{\psi_i}\}_{i=1}^M$ and sample next state $s_{h+1} = \widehat{T}(\cdot|s_h, a_h)$

18: $\quad\quad$ Add the model generated transition $(s_h, a_h, r_h, s_{h+1})$ into $\mathcal{D}_{\mathrm{model}}$

19: $\quad\quad$ Sample a batch of transitions from $\mathcal{D} \cup \mathcal{D}_{\mathrm{model}}$ with ratio $f$

20: $\quad\quad$ optimize $\pi_\theta$ with SAC: $\theta \leftarrow \theta - \lambda \nabla_\theta \mathcal{L}_{\mathrm{SAC}}(\theta)$

21: $\quad$ **end for**

22: $\quad$ **// Inner level update**

23: $\quad$ Compute loss $\mathcal{L}_{\mathrm{WSL}}(\psi, \nu) = -\mathbb{E}_{(s,a,s') \in \mathcal{D}} \left[ w_\nu(s,a,s') \log \left( \widehat{T}_\psi(s'|s,a) \right) \right]$

24: $\quad$ Perform gradient descent: $\psi \leftarrow \psi - \lambda \nabla_\psi \mathcal{L}_{\mathrm{WSL}}(\psi, \nu)$

25: $\quad$ **// Outer level update**

26: $\quad$ Compute $h$ according to Equation 10

27: $\quad$ Compute gradient $g_{\mathrm{RVL}} = h \cdot \nabla_\nu w_\nu(s,a,s')$

28: $\quad$ Perform gradient descent: $\nu \leftarrow \nu - \lambda \cdot g_{\mathrm{RVL}}$

29: **end for**

---

### D.3 HYPERPARAMETER TUNING OF ROMI

There are three main hyperparameters for ROMI: the uncertainty set scale $\xi$, the uncertainty set sampling number $N$, and the rollout length $H$. We show how we tune those hyperparameters below.

**Uncertainty set scale $\xi$.** For D4RL and NeoRL tasks, we search $\xi$ in the range of $\{0.01, 0.1, 1.0, 10\}$ and choose the value with the best performance.

Table 3: Base hyperparameter setup for ROMI shared across all runs.

| | Hyperparameter | Value |
|---|---|---|
| Dynamics Model | Ensemble size | 7 |
| | Architecture | (input, 200, 200, 200, 200, output) |
| | Real data ratio ($f$) | 0.5 |
| | Model learning batch size | 256 |
| | Model learning rate | $3 \times 10^{-4}$ |
| | Model pretraining epochs | 50 |
| Weighting network | Architecture | (input, 256, 256, 256, 1) |
| | Output range | [0.5, 2] |
| | learning rate | $1 \times 10^{-4}$ |
| | Batch size | 256 |
| Policy optimization | Require pretraining | False |
| | Critic learning rate | $3 \times 10^{-4}$ |
| | Actor learning rate | $1 \times 10^{-4}$ |
| | Optimizer | Adam (Kingma & Ba, 2014) |
| | Discount factor | 0.99 |
| | Soft update parameter | $5 \times 10^{-3}$ |
| | Target entropy | -dim($\mathcal{A}$) |
| | Batch size | 256 |

**Sampling number $N$.** $N$ controls the approximation error between the sampled minimum Q-value and the true minimum Q-value within the uncertainty set. A larger $N$ generally leads to a smaller error but incurs higher computational costs. Furthermore, a larger $\xi$ often necessitates a larger $N$. In our experiments, we select $N$ from the set $\{10, 20, 30\}$ for all the tasks. To conserve computational resources, if the performance difference between two candidate values of $N$ is not significant, we choose the smaller one.

**Rollout length $H$.** We perform short-horizon rollouts similar to previous works (Sun et al., 2023; Rigter et al., 2022), and tune $H$ across $\{2, 5\}$ for Gym and NeoRL tasks. For Antmaze tasks, we vary the parameters on $H \in \{1, 3\}$.

The selected hyperparameters for each task are listed in Table 4.

# E  COMPUTATIONAL COST

## E.1  COMPUTE INFRASTRUCTURE

We list our hardware specifications as follows:

- GPU: NVIDIA RTX A6000 ($\times 4$)
- CPU: AMD Ryzen Threadripper PRO 5975WX

We also list our software specifications as follows:

- Python: 3.8.18
- Pytorch: 1.11.0+cu113
- Tensorflow: 2.9.1
- Numpy: 1.24.4
- Gym: 0.22.0
- MuJoCo: 2.0
- D4RL: 1.1

Table 4: Hyperparameters of ROMI.

| Domain Name | Task Name | $\xi$ | $N$ | $H$ |
|---|---|---|---|---|
| D4RL MuJoCo | halfcheetah-random | 0.01 | 10 | 5 |
| | hopper-random | 0.1 | 10 | 5 |
| | walker2d-random | 0.01 | 10 | 5 |
| | halfcheetah-medium | 0.1 | 10 | 2 |
| | hopper-medium | 1.0 | 10 | 2 |
| | walker2d-medium | 0.01 | 10 | 5 |
| | halfcheetah-medium-replay | 0.1 | 10 | 5 |
| | hopper-medium-replay | 1.0 | 10 | 5 |
| | walker2d-medium-replay | 0.01 | 10 | 5 |
| | halfcheetah-medium-expert | 1.0 | 10 | 2 |
| | hopper-medium-expert | 1.0 | 10 | 2 |
| | walker2d-medium-expert | 0.01 | 10 | 5 |
| D4RL Antmaze | antmaze-umaze | 0.1 | 10 | 1 |
| | antmaze-umaze-diverse | 1.0 | 10 | 1 |
| | antmaze-medium-play | 0.1 | 10 | 3 |
| | antmaze-medium-diverse | 1.0 | 10 | 1 |
| | antmaze-large-play | 1.0 | 10 | 1 |
| | antmaze-large-diverse | 1.0 | 10 | 1 |
| NeoRL | HalfCheetah-L | 0.1 | 10 | 2 |
| | Hopper-L | 0.1 | 10 | 2 |
| | Walker2d-L | 1.0 | 10 | 2 |
| | HalfCheetah-M | 0.01 | 10 | 5 |
| | Hopper-M | 0.1 | 10 | 2 |
| | Walker2d-M | 0.01 | 10 | 2 |
| | HalfCheetah-H | 1.0 | 10 | 2 |
| | Hopper-H | 0.1 | 10 | 2 |
| | Walker2d-H | 0.01 | 10 | 5 |

### E.2 COMPUTING RESOURCE CONSUMPTION

We compare the computational resource consumption of ROMI and RAMBO, including model parameter size, GPU memory usage during training, and runtime, as shown in Table 5. The results indicate that while ROMI and RAMBO show no significant difference in model parameter size or GPU memory usage, ROMI requires more runtime. This is primarily due to the additional training overhead of the weighting network.

Table 5: Comparison of ROMI and RAMBO on computing resource consumption

| Task Name | Size of Parameters (MB) | | GPU Memory (GB) | | Runtime (s/epoch) | |
|---|---|---|---|---|---|---|
| | ROMI | RAMBO | ROMI | RAMBO | ROMI | RAMBO |
| hopper-random | 7.93 | 7.66 | 3.30 | 3.28 | 28.20 | 23.76 |
| halfcheetah-random | 8.19 | 7.91 | 3.33 | 3.32 | 29.41 | 22.24 |
| walker2d-random | 8.19 | 7.91 | 3.33 | 3.32 | 30.06 | 23.89 |
| hopper-medium | 7.93 | 7.66 | 3.30 | 3.28 | 28.43 | 23.31 |
| halfcheetah-medium | 8.19 | 7.91 | 3.33 | 3.32 | 29.91 | 22.63 |
| walker2d-medium | 8.19 | 7.91 | 3.33 | 3.32 | 29.67 | 23.15 |
| hopper-medium-replay | 7.93 | 7.66 | 3.30 | 3.28 | 28.46 | 22.27 |
| halfcheetah-medium-replay | 8.19 | 7.91 | 3.33 | 3.32 | 29.13 | 23.52 |
| walker2d-medium-replay | 8.19 | 7.91 | 3.33 | 3.32 | 29.64 | 23.11 |
| hopper-medium-expert | 7.93 | 7.66 | 3.30 | 3.28 | 28.33 | 22.93 |
| halfcheetah-medium-expert | 8.19 | 7.91 | 3.33 | 3.32 | 29.21 | 23.47 |
| walker2d-medium-expert | 8.19 | 7.91 | 3.33 | 3.32 | 30.02 | 23.09 |

## F MORE EXPERIMENTAL RESULTS

In this part, we provide more experimental results missing from the main text.

### F.1 D4RL ANTMAZE RESULTS

We compare ROMI with several model-based offline RL methods on the challenging D4RL Antmaze tasks; results are presented in Table 6. Performance varies substantially across methods: MOPO fails to achieve meaningful performance on all six tasks; COMBO performs well on `umaze` and `umaze-diverse` but fails on others; RAMBO achieves nonzero scores on four of the six tasks, but its total score remains low. MOBILE performs strongly overall, achieving nonzero scores on all tasks with a total of 173.4. In contrast, ROMI achieves a total score of **186.5**– higher than MOBILE and the highest among all compared methods. Notably, ROMI clearly outperforms MOBILE on `umaze-diverse` and `medium-diverse`, and exceeds RAMBO on five of the six tasks. The only exception is `large-play`, where both methods score zero. These results demonstrate ROMI's effectiveness in addressing challenging Antmaze domains.

### F.2 MORE RESULTS ON MOTIVATION EXAMPLE

We conduct experiments on nine D4RL datasets to examine the limitations of RAMBO. In Section 3, we presented results on the `halfcheetah-medium-expert-v2` dataset due to space constraints. Here, we provide results across all nine datasets, including gradient norms, normalized scores, and Q-value estimates.

Figure 6 shows the gradient norm of the adversarial loss during RAMBO training for $\lambda = 0.05$ and $\lambda = 0.1$. We omit curves for smaller $\lambda$ such as $\lambda = 3 \times 10^{-4}$ since we find the gradient norm stays small for those $\lambda$. as the gradient norm remains consistently small in those cases. As shown, gradient explosion occurs on two tasks (`hopper-medium-replay-v2` and `hopper-medium-expert-v2`) with $\lambda = 0.05$. For $\lambda = 0.1$, this issue becomes more frequent, occurring in 7 out of 9 tasks. These results demonstrate that RAMBO suffers from training instability unless $\lambda$ is set to an extremely small value.

Table 6: Successful Success rate comparison on Antmaze tasks. Results are averaged over 5 seeds. The top two scores for each task are **bold**.

| Task Name | MOPO | COMBO | RAMBO | MOBILE | ROMI (Ours) |
|---|---|---|---|---|---|
| antmaze-umaze | 0.0 | **80.3** | $42.6 \pm 11.9$ | **77.0** | $70.3\pm 11.4$ |
| antmaze-umaze-diverse | 0.0 | **57.3** | $12.9 \pm 5.6$ | 20.4 | $\mathbf{32.8} \pm 13.0$ |
| antmaze-medium-play | 0.0 | 0.0 | $18.7 \pm 9.2$ | **64.6** | $\mathbf{51.3} \pm 19.7$ |
| antmaze-medium-diverse | 0.0 | 0.0 | $\mathbf{20.4} \pm 3.7$ | 1.6 | $\mathbf{30.1} \pm 10.0$ |
| antmaze-large-play | 0.0 | 0.0 | $0.0 \pm 0.0$ | **2.6** | $0.0 \pm 0.0$ |
| antmaze-large-diverse | 0.0 | 0.0 | $0.0 \pm 0.0$ | **7.2** | $\mathbf{6.4} \pm 4.2$ |
| Total | 0.0 | 137.6 | 94.6 | **173.4** | **186.5** |

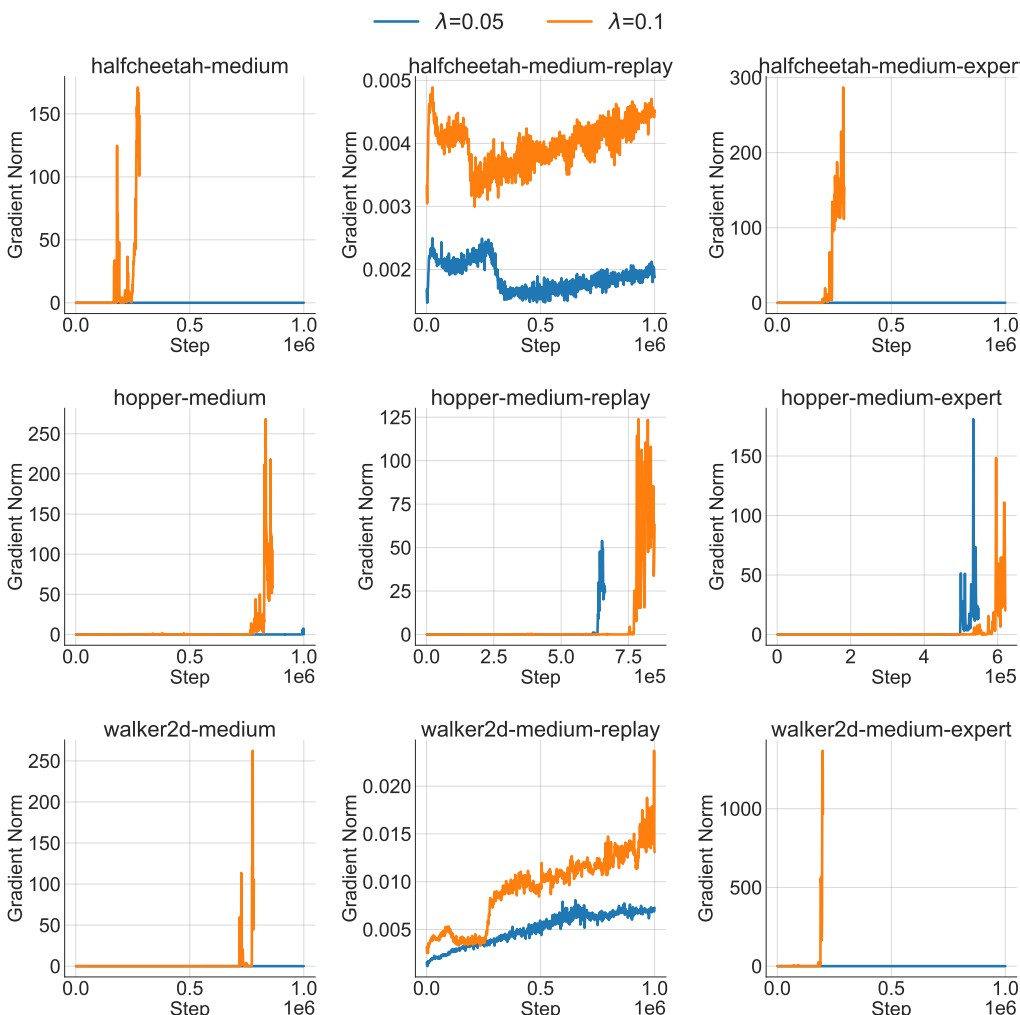

Figure 6: Gradient norm curves on 9 D4RL datasets for RAMBO when $\lambda = 0.05$ and $\lambda = 0.1$

Figure 7 and Figure 8 present the normalized score and Q-value estimation on 9 D4RL datasets for RAMBO across different $\lambda$. A clear trend can be observed: setting $\lambda$ to very small values within $\{0, 3e-4, 5e-3\}$ has almost no impact on performance and Q-value estimation. In contrast, a larger value of $\lambda$ such as 0.05 and 0.1 leads to diverged Q-value and performance collapse, indicating that RAMBO is highly sensitive to $\lambda$ and prone to excessive conservatism.

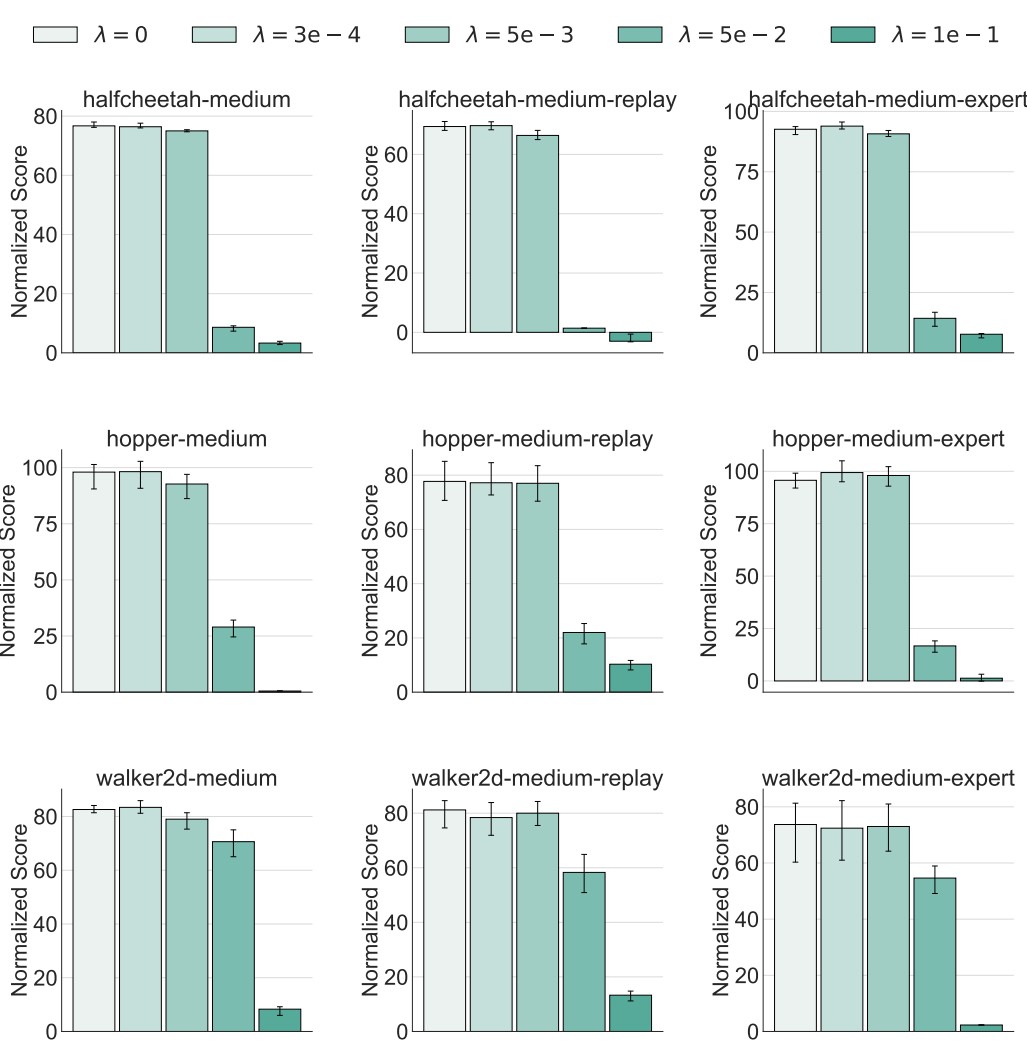

Figure 7: Normalized score comparison on 9 D4RL datasets across different $\lambda$ for RAMBO.

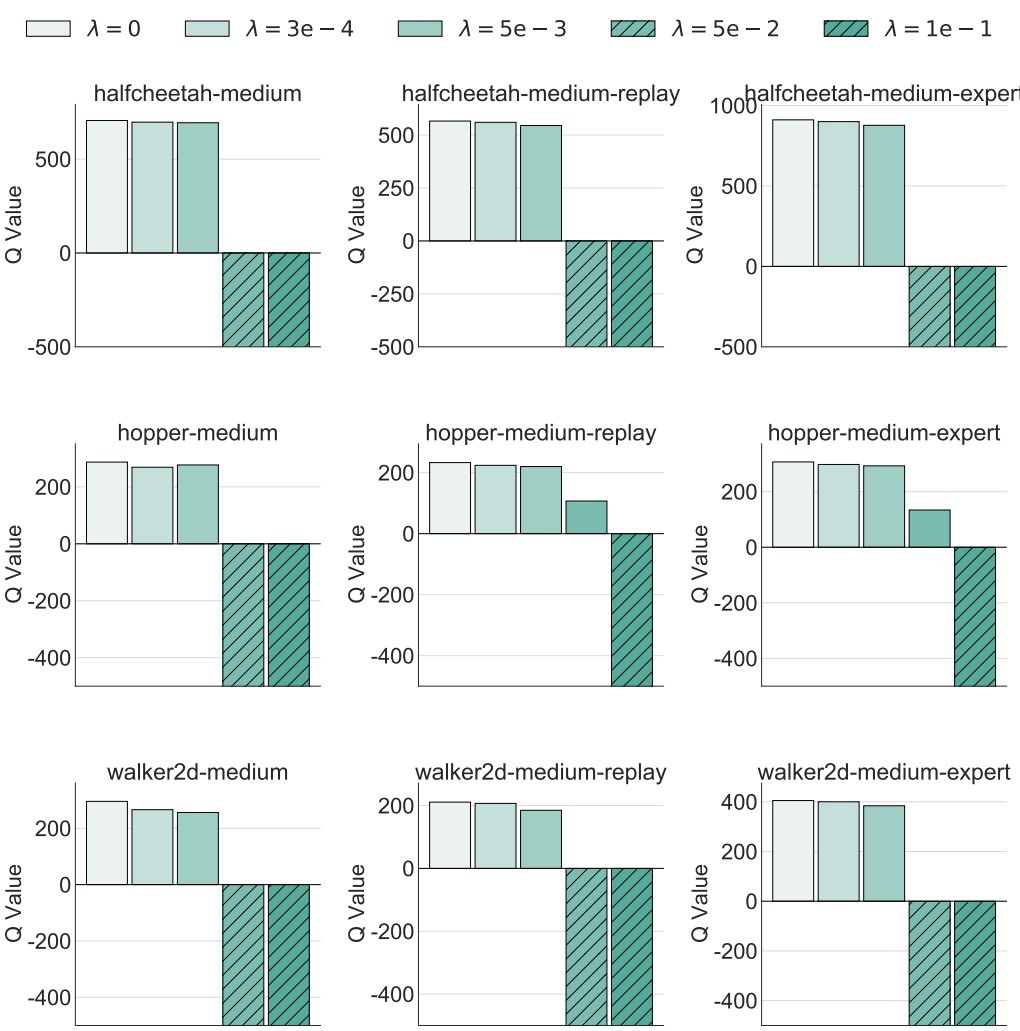

Figure 8: Q-value estimation on 9 D4RL datasets across different $\lambda$ for RAMBO.

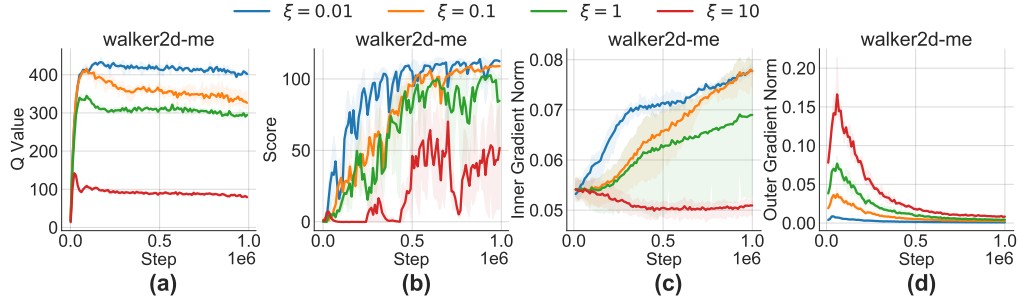

Figure 9: Comparison for different $\xi$ in terms of Q-value, normalized score, inner loss gradient norm, outer loss gradient norm.

Table 7: Performance comparison for different $N$ on 6 D4RL datasets.

| Datasets | $N = 10$ | $N = 20$ | $N = 30$ |
|---|---|---|---|
| half-mr | 70.7±4.8 | 71.8±4.2 | 72.2±5.1 |
| hopper-mr | 102±2.4 | 100.0±1.5 | 103.3±2.8 |
| walker2d-mr | 85.2±2.6 | 87.4± 4.6 | 81.5 ± 3.8 |
| half-me | 108.5±4.2 | 110.4±5.3 | 111.0±3.4 |
| hopper-me | 111.5±3.8 | 110.0±1.4 | 113.9± 4.7 |
| walker2d-me | 113.3±1.9 | 115.6±2.8 | 111.7±0.5 |
| Average | 98.5 | 99.2 | 99.1 |

### F.3 MORE PARAMETER STUDY

In this section, we further study the sensitivity of ROMI to the choices of the uncertainty set scale $\xi$ and sampling number $N$.

We present the parameter study results of $\xi$ on the `halfcheetah-medium-expert-v2` dataset in Section 5.3. Here we present more results on `walker2d-medium-expert-v2` dataset. We vary $\xi$ across $\{0.01, 0.1, 1.0, 10\}$ and compare the normalized score, Q-value estimate and the gradient norm of the inner and outer loss. The results are shown in Figure 9. We can see that no severe Q-value underestimation or gradient explosion occurs across different $\xi$, verifying the superiority of ROMI over RAMBO.

We also study the sensitivity of $N$. We sweep $N$ across $\{10, 20, 30\}$ for each task, and present the results on 6 D4RL datasets in Table 7. We find that setting $N$ to 20 and 30 does not show clear performance improvement compared to $N = 10$. For the consideration of computational cost, we fix $N = 10$ for all the experiments without further tuning.

### F.4 LEARNING CURVES

We provide the learning curves of RAMBO, MOBILE, and ROMI on 12 D4RL datasets in Figure 10 and Figure 11. The training steps are 1M for all the methods.

## G LLM USAGE STATEMENT

In this work, we use the large language model (LLM) to polish the grammar of our manuscript. However, we confirm that no LLMs are involved in the key steps of this work, including the conceptualization of the idea, the development of methodology, and theoretical proofs.

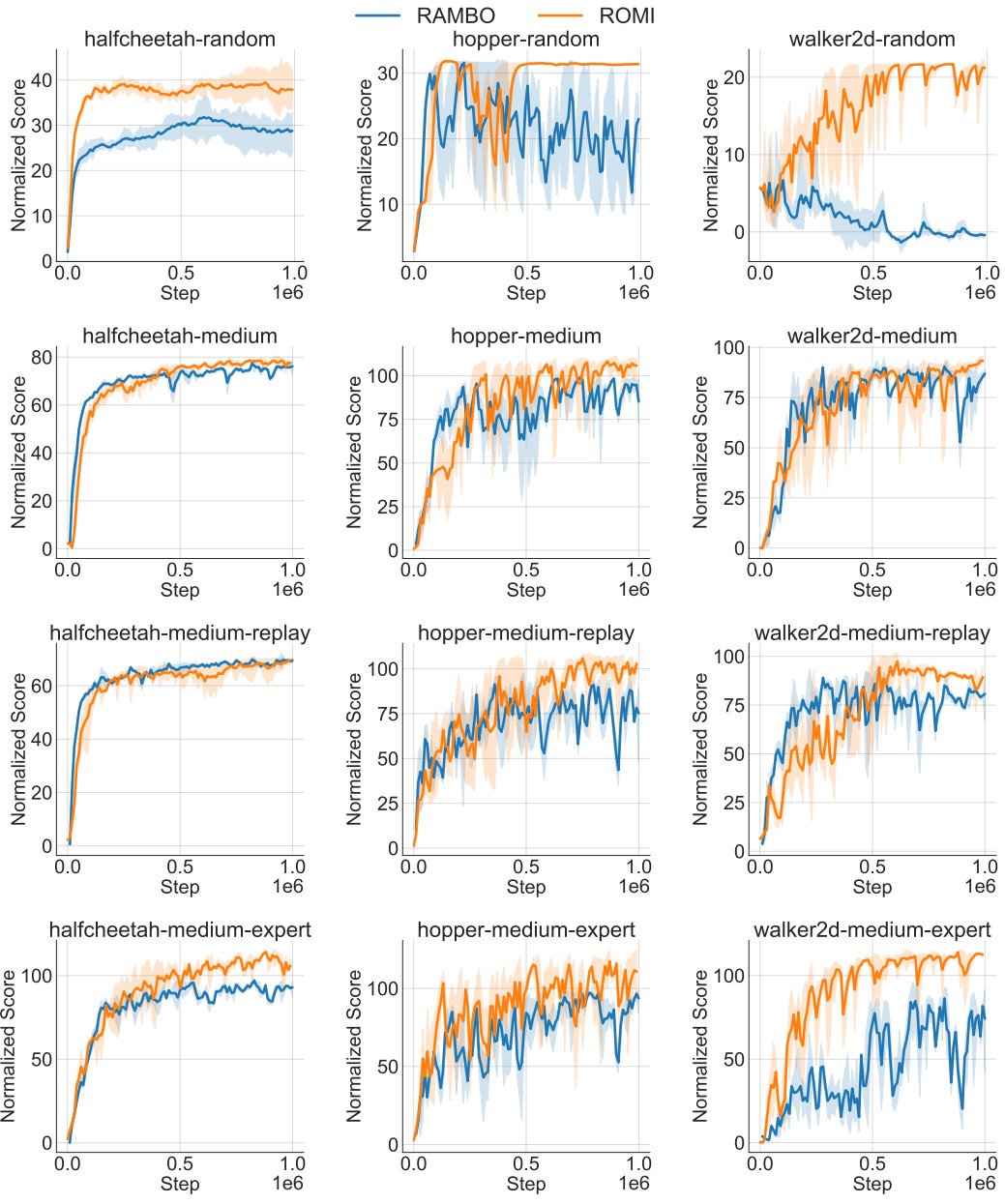

Figure 10: Learning curves of RAMBO and ROMI on 12 D4RL datasets. Averaged over 5 seeds.

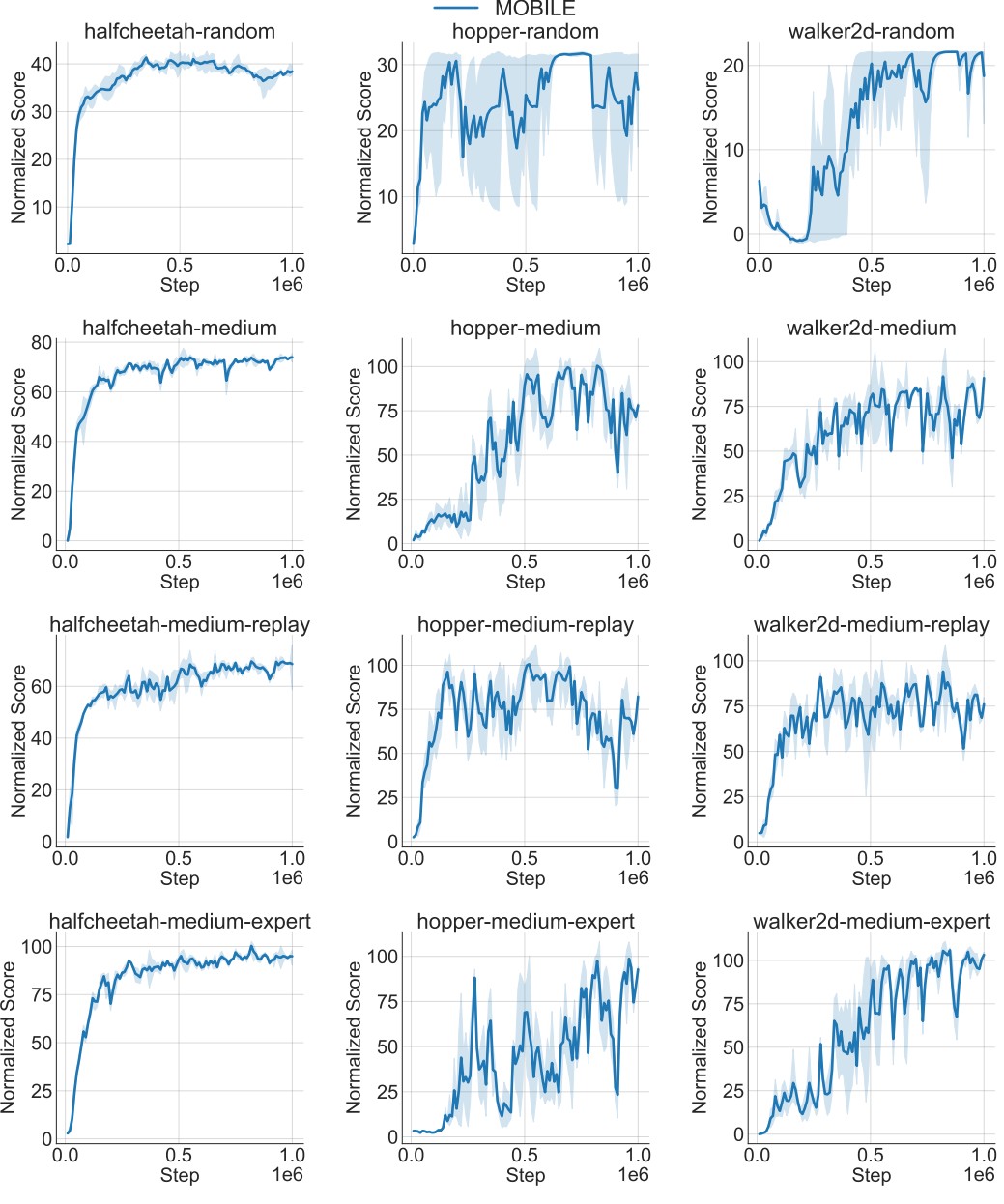

Figure 11: Learning curves of MOBILE on 12 D4RL datasets. Averaged over 5 seeds.

