# OpenReview forum: "Model-based Offline RL via Robust Value-Aware Model Learning with Implicitly Differentiable Adaptive Weighting"
_ICLR.cc/2026/Conference — ICLR 2026 Poster_

### Official Review · Reviewer_VuTF · 2025-10-27

**Soundness:** 3
**Presentation:** 3
**Contribution:** 3
**Rating:** 8
**Confidence:** 4

**Summary:**

The paper proposes an alternative approach to model-based offline reinforcement learning, building on the idea of value-aware model learning. A model is constructed that minimizes an adversarial value difference error as a weighing to a log likelihood reconstruction loss. The authors show the efficacy of their method on standard offline RL tasks, in addition to providing some theoretical insights into the method.

**Strengths:**

While the overall method is slightly complex, the paper presents the motivation and contribution of the work clearly. The proposed integration of conservative value-aware model learning into offline model-based RL is reasonable and shows some promising results. In addition, known issues about the generalization abilities of value-aware model learning are appropriately avoided with the bi-level optimization scheme.

**Weaknesses:**

I genuinely do not have any major issues with the paper, so the following are excessively nitpicky and mostly comments on the writing.

I believe the writing of the paper could be strengthened slightly by defining the method less in contrast to RAMBO, as I believe the method conceptually stands on it's own.

Line 175: I don't believe this is a proper example of an adversarial loss, as there is no min/max formulation with competing networks.

It would be polite to cite Farahmand for coining the term value-aware as soon as it appears. Since the Farahmand paper is cited, I don't believe this is an egregious error. Similarly, the method is somewhat conceptually similar to Voelcker et al., as a reweighted regular MLE loss is considered. Again, prior work is acknowledged, so this is not really an issue.

**Questions:**

How is the state distance metric defined? This seems to be a crucial implementation detail.

How are the uncertainty intervals in the empirical section computed? Are bolded results statistically significant? Why do aggregated results and baseline lack uncertainty intervals?

Can the method work for higher dimensional problems as well? Pessimism over a state uncertainty set could conceivably be harder to tune for larger more high-dimensional problems, and all those tested here are rather low-dimensional.

How was hyperparameter tuning and model selection conducted? Was there a separate offline validation set, or where hyperparameters tuned by testing in the online environment.

---

> ### Author Response · Authors · 2025-11-23
> **Reply to Reviewer VuTF (Part 1)**
>
> We sincerely appreciate the reviewer for reviewing our paper and for acknowledging our contribution. We present our responses to the concerns below. We hope our responses could address the reviewer's concerns.
>
> ## Concern 1: ROMI could be defined less in contrast to RAMBO
>
> We deeply appreciate this suggestion! ROMI could certainly be defined on its own. However, since ROMI is motivated by RAMBO's limitations, we believe introducing ROMI in contrast to RAMBO could help readers more clearly understand our motivation and contribution.
>
> ## Concern 2: Line 175 is not a proper example of adversarial loss
>
> Thanks for pointing this out. We agree that it is more accurately characterized as a value-aware model loss than an adversarial loss. We have corrected this in the revision.
>
> ## Concern 3: Several previous works can be cited as soon as they appear
>
> We appreciate the reviewer's suggestion. We have moved the citations of Farahmand [1] and Voelcker [2] to an earlier position in the **Introduction**​ section in our revision.
>
> ## Concern 4: How is the state distance metric defined?
>
> The state distance metric is defined as the Euclidean distance. Specifically, for any two states $s_1$ and $s_2$, the distance is computed as $d(s _ 1,s _ 2)=\\|s _ 1-s _ 2\\| _ 2$.

---

> > ### Author Response · Authors · 2025-11-23
> > **Reply to Reviewer VuTF (Part 2)**
> >
> > ## Concern 5: How are the uncertainty intervals computed?
> >
> > We appreciate this question. The uncertainty interval is computed as the standard deviation of the performance over 1M steps across 5 random seeds for each method. Due to table width limitations, we only present the uncertainty interval for our method (ROMI) in Table 1 and Table 2. The full results, including the intervals for other methods, are provided in Appendix F.4 in our revision and below.
> >
> > In Table 1 and Table 2, we initially bolded only the best result for each dataset. It is possible that a second-best method had a similar performance but was not bolded. We have rectified this issue by bolding all best results with similar performance in our revision.
> >
> > **D4RL**
> >
> > | Dataset | CQL | IQL | MOPO | RAMBO | COUNT | MOBILE | ROMI |
> > |-|-|-|-|-|-|-|-|
> > | hopper-r | 7.9 $\pm$ 2.5 | 7.6 $\pm$ 0.8 | 27.8 $\pm$ 4.8 | 23.9 $\pm$ 4.4 | **30.0 $\pm$ 1.8** | 28.5 $\pm$ 3.8 | **31.7 $\pm$ 0.3** |
> > | halfcheetah-r | 17.5 $\pm$ 3.5 | 13.1 $\pm$ 2.6 | 36.7 $\pm$ 1.1 | 29.5 $\pm$ 3.7 | 37.3 $\pm$ 3.1 | **39.4 $\pm$ 1.3** | **38.4 $\pm$ 4.1** |
> > | walker2d-r | 5.1 $\pm$ 0.5 | 5.4 $\pm$ 0.3 | 0.0 $\pm$ 0.0 | 0.0 $\pm$ 0.0 | 19.9 $\pm$ 1.2 | **23.6 $\pm$ 2.1** | **22.4 $\pm$ 2.6** |
> > | hopper-m | 53.0 $\pm$ 1.9 | 66.2 $\pm$ 1.3 | 72.2 $\pm$ 4.7 | 95.2 $\pm$ 5.7 | 100.2 $\pm$ 4.7 | 77.6 $\pm$ 1.5 | **105.0 $\pm$ 3.3** |
> > | halfcheetah-m | 47.0 $\pm$ 0.2 | 47.9 $\pm$ 0.1 | 64.0 $\pm$ 3.1 | **76.2 $\pm$ 0.4** | **75.9 $\pm$ 0.3** | 74.6 $\pm$ 0.3 | **76.2 $\pm$ 0.6** |
> > | walker2d-m | 73.4 $\pm$ 2.5 | 67.2 $\pm$ 1.0 | 82.1 $\pm$ 4.7 | 83.9 $\pm$ 3.2 | 88.4 $\pm$ 4.2 | **90.7 $\pm$ 1.0** | **93.6 $\pm$ 1.3** |
> > | hopper-mr | 88.7 $\pm$ 3.5 | 94.0 $\pm$ 2.1 | 90.5 $\pm$ 4.4 | 77.2 $\pm$ 8.3 | 97.6 $\pm$ 3.3 | 85.7 $\pm$ 0.9 | **102.0 $\pm$ 2.4** |
> > | halfcheetah-mr | 45.3 $\pm$ 0.2 | 44.3 $\pm$ 0.2 | 52.1 $\pm$ 1.4 | **69.7 $\pm$ 0.8** | **70.5 $\pm$ 1.0** | 67.7 $\pm$ 1.2 | **70.7 $\pm$ 1.8** |
> > | walker2d-mr | 81.8 $\pm$ 3.9 | 74.8 $\pm$ 2.0 | 73.6 $\pm$ 1.5 | 81.2 $\pm$ 6.1 | **87.5 $\pm$ 2.0** | 75.0 $\pm$ 0.7 | 85.2 $\pm$ 3.6 |
> > | hopper-me | 105.9 $\pm$ 2.8 | 92.7 $\pm$ 2.2 | 84.5 $\pm$ 3.0 | 99.7 $\pm$ 3.6 | **109.8 $\pm$ 7.4** | 95.6 $\pm$ 3.2 | **110.5 $\pm$ 6.8** |
> > | halfcheetah-me | 75.6 $\pm$ 1.5 | 86.3 $\pm$ 1.7 | 92.3 $\pm$ 3.8 | 93.9 $\pm$ 1.5 | 100.0 $\pm$ 1.3 | 96.1 $\pm$ 1.4 | **104.5 $\pm$ 2.2** |
> > | walker2d-me | 107.9 $\pm$ 0.5 | **110.7 $\pm$ 1.4** | 93.8 $\pm$ 3.7 | 73.7 $\pm$ 8.9 | **110.4 $\pm$ 1.6** | 103.2 $\pm$ 1.2 | **113.3 $\pm$ 1.9** |
> > | Total | 709.1 | 710.2 | 769.6 | 804.1 | 927.5 | 857.7 | **953.5** |
> >
> >
> >
> >
> > **NeoRL**
> >
> >
> > | Dataset | CQL | IQL | EDAC | MOPO | RAMBO | MOBILE | ROMI |
> > |-|-|-|-|-|-|-|-|
> > | Hopper-L | 16.8 $\pm$ 0.4 | 14.3 $\pm$ 0.9 | 12.2 $\pm$ 0.2 | 12.0 $\pm$ 0.4 | 15.1 $\pm$ 0.2 | 18.1 $\pm$ 0.3 | **22.4 $\pm$ 0.5** |
> > | HalfCheetah-L | 35.0 $\pm$ 1.1 | 34.1 $\pm$ 1.6 | 31.3 $\pm$ 1.4 | 34.2 $\pm$ 3.5 | 30.1 $\pm$ 5.3 | **38.9 $\pm$ 4.0** | 35.8 $\pm$ 3.9 |
> > | Walker2d-L | **42.4 $\pm$ 2.1** | 40.1 $\pm$ 2.6 | **43.1 $\pm$ 1.5** | 15.6 $\pm$ 1.7 | 24.3 $\pm$ 3.0 | 39.0 $\pm$ 2.5 | 36.4 $\pm$ 2.3 |
> > | Hopper-M | **58.1 $\pm$ 4.6** | **56.8 $\pm$ 2.1** | 37.2 $\pm$ 4.2 | 0.0 | 37.5 $\pm$ 3.1 | 47.3 $\pm$ 7.0 | 46.6 $\pm$ 6.8 |
> > | HalfCheetah-M | 50.0 $\pm$ 1.0 | 52.0 $\pm$ 0.6 | 54.6 $\pm$ 3.5 | 53.1 $\pm$ 2.6 | 51.2 $\pm$ 1.9 | **55.6 $\pm$ 1.2** | **57.7 $\pm$ 1.4** |
> > | Walker2d-M | 52.5 $\pm$ 2.4 | 51.2 $\pm$ 1.9 | **53.9 $\pm$ 0.6** | 35.7 $\pm$ 3.4 | 36.2 $\pm$ 4.5 | 51.4 $\pm$ 1.7 | **54.9 $\pm$ 2.0** |
> > | Hopper-H | **65.6 $\pm$ 3.0** | 62.4 $\pm$ 2.1 | 56.7 $\pm$ 1.8 | 17.9 $\pm$ 3.3 | 48.7 $\pm$ 5.5 | **65.0 $\pm$ 4.7** | **65.9 $\pm$ 4.5** |
> > | HalfCheetah-H | 74.6 $\pm$ 1.0 | **76.3 $\pm$ 1.8** | 72.0 $\pm$ 2.9 | 63.2 $\pm$ 2.3 | 72.3 $\pm$ 3.5 | 69.4 $\pm$ 3.4 | **77.4 $\pm$ 2.7** |
> > | Walker2d-H | 71.3 $\pm$ 1.5 | **74.0 $\pm$ 0.9** | 73.1 $\pm$ 2.0 | 26.4 $\pm$ 5.2 | 67.4 $\pm$ 4.3 | 71.7 $\pm$ 1.0 | **75.1 $\pm$ 1.8** |
> > | Total | 466.3 | 461.2 | 434.1 | 258.1 | 382.8 | 456.4 | **472.2** |

---

> > > ### Comment · Reviewer_VuTF · 2025-11-24
> > > **Statistical nitpicking**
> > >
> > > One standard deviation is a relatively meaningless measure of variation and it is not quite clear how it should be interpreted. In the interest of scientific accuracy etc., I strongly encourage you to look into better reporting on confidence intervals, e.g. Agarwal's "Edge of the statistical precipice" or Patterson's "Empirical Design in RL"
> > >
> > > [1] https://agarwl.github.io/rliable/
> > >
> > > [2] https://jmlr.org/papers/volume25/23-0183/23-0183.pdf

---

> ### Author Response · Authors · 2025-11-23
> **Reply to Reviewer VuTF (Part 3)**
>
> ## Concern 6: Can the method work for higher dimensional problems as well?
>
> We appreciate the suggestion of testing ROMI on high-dimensional tasks. To address this concern, we evaluate ROMI on the Adroit domain, which involves long-horizon, sparse-reward and high-dimensional robotics manipulation tasks, and is more challenging than MuJoCo for offline RL. We compare ROMI against several baselines (CQL, IQL, MOPO, RAMBO, MOBILE) on six Adroit datasets. The results are presented below.
>
> |Dataset| CQL | IQL | MOPO | RAMBO | MOBILE | ROMI|
> |-|-|-|-|-|-|-|
> |pen-human|36.5$\pm$5.7|54.8$\pm$8.6|10.2$\pm$4.3|13.5$\pm$5.1|36.7$\pm$7.6|**63.8$\pm$8.4**|
> |door-human|**9.9$\pm$4.3**|3.4$\pm$1.4|0.0|0.0|0.0|0.0|
> |hammer-human|**4.1$\pm$2.0**|2.2$\pm$0.6|0.0|1.1$\pm$0.2|1.2$\pm$0.5|2.2$\pm$0.2|
> |pen-cloned|39.2$\pm$6.6|35.1$\pm$7.5|42.7$\pm$8.8|33.7$\pm$11.3|**51.8$\pm$8.6**|**57.7$\pm$3.9**|
> |door-cloned|0.5$\pm$0.2|0.6$\pm$0.5|12.9$\pm$4.2|13.6$\pm$6.2|**16.2$\pm$6.4**|**14.3$\pm$5.8**|
> |hammer-cloned|**2.0$\pm$0.4**|1.5$\pm$1.0|0.0|0.0|0.0|0.0|
> |Total|92.2|97.6|65.8|61.9|105.9|**138.0**|
>
> The results show that ROMI achieves superior performance on several datasets (e.g., pen-human, pen-cloned) and obtains the highest total score (138.0) across all six datasets. However, we also observe that ROMI, along with other model-based offline methods, struggles on the door-human and hammer-cloned datasets. This is likely due to the difficulty of learning an accurate dynamics model from these datasets. It would be interesting to develop more effective algorithms for these high-dimensional tasks for future work.
>
> ## Concern 7: How was hyperparameter tuning conducted?
>
> We search the hyperparameters across candidate sets and run parallel experiments with online evaluation using the Ray framework [3], selecting the hyperparameter values with the best performance. This online tuning method is widely adopted in previous offline RL studies.
> However, an offline hyperparameter selection method could also be applied to ROMI, similar to RAMBO. Specifically, we could define a heuristic for hyperparameter selection, such as $\min(Q_\mathrm{avg}+Q_\mathrm{var})$, as demonstrated in Appendix B.5 in RAMBO paper.
>
> [1] Value-aware loss function for model-based reinforcement learning. AISTATS 2017
>
> [2] Value gradient weighted model-based reinforcement learning. ICLR 2022
>
> [3] Ray: A Distributed Framework for Emerging AI Applications. 2017

---

> > ### Comment · Reviewer_VuTF · 2025-11-24
> > **Thank you**
> >
> > I thank the authors for addressing my comments. I have some gripes about the statistical methodology, as pointed out above, and I do think that the offline RL field has some fundamental issues with hyperparameter tuning (which shouldn't be done in online rollouts, otherwise the offline paradigm is pointless). As these are pervasive issues in this field and not a fault of the authors, I simply ask you to reflect on these and think about better research methodology.
> >
> > I keep my recommendation for acceptance.

---

> > > ### Author Response · Authors · 2025-11-25
> > > **Follow up with Reviewer VuTF**
> > >
> > > We are grateful to the reviewer for the insightful suggestion regarding statistical significance. We will explore better offline hyperparameter tuning methods in our future work. Once again, we thank the reviewer for taking time to review our paper and for the recognition of our work.

---

### Official Review · Reviewer_rQhm · 2025-10-29

**Soundness:** 3
**Presentation:** 3
**Contribution:** 3
**Rating:** 8
**Confidence:** 4

**Summary:**

This paper introduces ROMI, a new method for model-based offline reinforcement learning (RL) that addresses instability and over-conservatism found in previous approaches like RAMBO. ROMI uses a robust value-aware model learning strategy and a bi-level optimization scheme to better control conservatism and improve generalization. Experiments show ROMI outperforms RAMBO and matches or exceeds other state-of-the-art methods on standard offline RL benchmarks.

**Strengths:**

- Proposes a novel, robust approach that addresses instability and over-conservatism in model-based offline RL. The bi-level optimization scheme appears a promising solution for adaptive conservatism, improving both stability and performance
- Demonstrate strong empirical results on MuJoCo datasets, outperforming or matching state-of-the-art baselines
- Proposed framework appears to be less sensitive to hyperparameters and more generalizable across tasks, as shown e.g. in Fig. 3

**Weaknesses:**

- Even though robustness has clearly gone up, the approach may still require careful tuning of certain parameters - i.e. in Fig. 3 we see that while 0.01-1 are close together, setting the parameter to 10 yields very unstable results. Since all evaluations are in a relatively similar task domain (MuJoCo hopper, walker and halfcheetah), it is unclear whether the range of stable parameters is representable for other tasks as well, i.e. for other environments tuning the parameter may again become an issue.
- More generally speaking, while the authors clearly demonstrate performance gains on a set of well established offline RL benchmarks, the distribution of these tasks is quite narrow - it would be interesting to see how the algorithm performs on a broader selection
- Another downside I see is, if I am not mistaken, that the degree of uncertainty that governs the training needs to be set a priori and cannot be adjusted at runtime. Prior methods, such as [1,2,3,4] (could also help your related work section) have developed policies that can be adapted immediately when it is observed that the behavior is too far OOD or too conservative, which can be an advantage over having to go back and retrain. Of course, this point does not necessarily have to be addressed, the proposed method has other demonstrated merits which justify publication.
- The method introduces additional complexity through the bi-level optimization, which may increase implementation difficulty and be a burden in practical application

[1] Ghosh, D., Ajay, A., Agrawal, P., & Levine, S. Offline RL Policies Should be Trained to be Adaptive. Proceedings of the 39th International Conference on Machine Learning (ICML), 2022.

[2] Hong, J., Kumar, A., & Levine, S. Confidence-Conditioned Value Functions for Offline Reinforcement Learning. International Conference on Learning Representations (ICLR), 2023.

[3] Swazinna, P., Udluft, S., & Runkler, T. User Interactive Offline Reinforcement Learning. International Conference on Learning Representations (ICLR), 2023.

[4] Zhang, Y., Liu, J., & Wang, Y. Train Once, Get a Family: State-Adaptive Balances for Offline-to-Online Reinforcement Learning. Advances in Neural Information Processing Systems (NeurIPS), 2023.

**Questions:**

Am I correctly assuming that the parameter governing the size of the uncertainty set has to be set a priori?

---

> ### Author Response · Authors · 2025-11-23
> **Reply to Reviewer rQhm**
>
> We thank the reviewer for the thoughtful review, and we sincerely appreciate that the reviewer acknowledges the contribution of our work. Below, we will address the raised concerns in detail, and we hope our responses can adequately resolve the reviewer's concerns.
>
> ## Concern 1: The requirement of tuning hyperparameters across different tasks
>
> We appreciate this concern raised by the reviewer. Yes, the value of $\xi$ needs to be tuned, and the optimal hyperparameter may vary for different tasks. However, this is also true for other SOTA model-based offline RL methods such as MOBILE and Count-MORL, which require tuning several critical hyperparameters to achieve the best performance.
>
> Moreover, although the results in Figure 3 indicate that performance degrades for a large $\xi$, this still demonstrates the key advantage of our method over RAMBO: it allows for a wider range of hyperparameter tuning to control conservatism, without causing Q-value divergence or gradient explosion. In contrast, RAMBO must limit the adversarial coefficient $\lambda$ to an extremely small value, making effective tuning nearly impossible.
>
>
> ## Concern 2: The distribution of selected tasks is narrow
>
> We appreciate the suggestion of testing ROMI on more tasks in addition to MuJoCo. To address this concern, we have conducted additional experiments on the Adroit domain, which features long-horizon, sparse-reward, high-dimensional robotic manipulation tasks that pose a significant challenge for offline RL algorithms. We compare ROMI against several baselines (CQL, IQL, MOPO, RAMBO, MOBILE) on six Adroit datasets. The results are presented below.
>
> |Dataset| CQL | IQL | MOPO | RAMBO | MOBILE | ROMI|
> |-|-|-|-|-|-|-|
> |pen-human|36.5$\pm$5.7|54.8$\pm$8.6|10.2$\pm$4.3|13.5$\pm$5.1|36.7$\pm$7.6|**63.8$\pm$8.4**|
> |door-human|**9.9$\pm$4.3**|3.4$\pm$1.4|0.0|0.0|0.0|0.0|
> |hammer-human|**4.1$\pm$2.0**|2.2$\pm$0.6|0.0|1.1$\pm$0.2|1.2$\pm$0.5|2.2$\pm$0.2|
> |pen-cloned|39.2$\pm$6.6|35.1$\pm$7.5|42.7$\pm$8.8|33.7$\pm$11.3|**51.8$\pm$8.6**|**57.7$\pm$3.9**|
> |door-cloned|0.5$\pm$0.2|0.6$\pm$0.5|12.9$\pm$4.2|13.6$\pm$6.2|**16.2$\pm$6.4**|**14.3$\pm$5.8**|
> |hammer-cloned|**2.0$\pm$0.4**|1.5$\pm$1.0|0.0|0.0|0.0|0.0|
> |Total|92.2|97.6|65.8|61.9|105.9|**138.0**|
>
> **The results show that ROMI achieves superior performance on several datasets (e.g., pen-human, pen-cloned) and obtains the highest total score (138.0) across all six datasets**. However, we also observe that ROMI, along with other model-based offline methods, struggles on the door-human and hammer-cloned datasets. This is likely due to the difficulty of learning an accurate dynamics model from these datasets. It would be interesting to develop more effective algorithms for these datasets in the future.
>
>
> ## Concern 3: The degree of uncertainty needs to be set a priori and cannot be adjusted at runtime
>
> Yes, $\xi$ must be specified before training. This occurs in prior methods such as RAMBO and MOBILE. The key distinction is that $\xi$ **offers more precise and direct control over the degree of conservatism** compared to an adversarial or penalty coefficient.
>
> We agree that the runtime adjustment of conservatism is quite promising. We thank the reviewer for highlighting this direction and for the related references, which we have now discussed in our limitation section. It would be interesting to explore how to adaptively adjust the conservatism in the future.
>
> ## Concern 4: The implementation complexity of the bi-level optimization
>
> The bi-level optimization framework introduces a slightly higher, but not substantially larger, implementation complexity compared to RAMBO. Specifically, the inner level minimizes an MLE loss weighted by $w_\nu$, which is implemented similarly to the standard MLE loss used in RAMBO. The outer level uses implicit differentiation to optimize the robust value-aware model loss, which can also be conveniently implemented with automatic differentiation in PyTorch.

---

> > ### Author Response · Authors · 2025-11-28
> >
> > Dear Reviewer rQhm, thank you very much for the helpful review and positive rating of our work. It would be great if you could give us some comments on our rebuttal, and kindly check our revision.

---

### Official Review · Reviewer_7d1B · 2025-10-29

**Soundness:** 2
**Presentation:** 2
**Contribution:** 2
**Rating:** 4
**Confidence:** 3

**Summary:**

This paper looks into an instability noticed in RAMBO with underestimated Q-values and proposes a new method called ROMI to fix this. Their method aims to prevent model exploitation by training the model to balance predicting a close-by state with the lowest value, as well as getting the dynamics correct.

**Strengths:**

Interesting finding in RAMBO, seemingly strong results, includes ablations, and an interesting method of balancing competing objectives that I have not come across in RL before.

**Weaknesses:**

**W1.** There are lots of moving parts. It seems significantly more complicated to implement and understand than RAMBO.

**W2.** The authors say they compare all methods at 1M steps for fairness. However, since some baselines like MOBILE were tuned for 3M steps, it would be important to see if ROMI's performance holds up relative to MOBILE etc when trained for the full 3M steps.

**W3.** Larger compute requirement.

**W4.** They solve the problem of tuning lambda, but introduce more new hyperparameters: xi, H, N.

**Questions:**

**Q1.** Where are the results from for each method for both the D4RL and Neo-RL benchmarks? If they are from the authors own implementations, how did they ensure they were implemented correctly and tuned fairly?

**Q2.** How was Wasserstein distance chosen. Did the authors try or consider any other distance metrics?

---

> ### Author Response · Authors · 2025-11-23
> **Reply to Reviewer 7d1B (Part 1)**
>
> We thank the reviewer for the thoughtful review and for acknowledging the strengths of our work. Below, we will address the raised concerns in detail, and we hope our responses can adequately resolve the reviewer's concerns.
>
> ## Concern 1: ROMI is more complicated to implement and understand than RAMBO
>
> - **Regarding implementation:​** ROMI does not introduce significant implementation complexity. Its core component for model learning is a bi-level optimization framework. The inner level minimizes an MLE loss reweighted by $w_\nu(s,a,s^\prime)$, which is straightforward to implement. The outer level optimizes the robust value-aware model loss via implicit differentiation, which can be conveniently implemented in PyTorch. Specifically, after computing has shown in Appendix A, we directly minimize the loss function $\mathcal{L}=h\cdot w_\nu$ using automatic differentiation.
>
> - **Regarding understanding:** In essence, both RAMBO and ROMI aim to solve the following optimization problem
> $$
> \pi = \arg\max\_{\pi \in \Pi} \min\_{\hat{T} \in \mathcal{M}_{\xi}} V\_{\hat{T}}^{\pi}
> \quad \text{s.t.} \quad
> \mathcal{M}\_{\xi}
> = \big\\{ \hat{T} \  | \ \mathbb{E}\_{(s,a)\sim\mathcal{D}}
> \big[
> \mathrm{Dis}\big(\hat{T}\_{\mathrm{MLE}}(\cdot \mid s,a),\, \hat{T}(\cdot \mid s,a)\big)
> \big]
> \leq \xi
> \big\\},
> $$
> which serves as a theoretical framework for model-based offline RL, as shown in Equation (1) in our paper. In practice, as discussed in Equation (2) in our paper, RAMBO proposed a simplified primal-dual reformulation for estimating the solution to this optimization problem, **which nevertheless results in difficulties in controlling the conservatism level and can be unstable during training**. To resolve such challenging issues, **ROMI proposes a different but practical reformulation of this problem by introducing two core components: robust value-aware model learning and implicitly differentiable adaptive weighting, which naturally leads to a bi-level optimization problem** with an improved empirical performance. Thus, the mathematical form of our method is elaborate, but the underlying motivation and intuition behind our method remain clear and easy to understand. We believe our method also provides a valuable insight into better solving the model-based offline RL problem practically.

---

> ### Author Response · Authors · 2025-11-23
> **Reply to Reviewer 7d1B (Part 2)**
>
> ## Concern 2: The performance comparison for a training step of 3M
>
> We appreciate the reviewer's suggestion and evaluate our method and other baselines under 3M training steps on D4RL and NeoRL datasets. We present the performance comparison below. The results indicate that, **even under 3M training steps, ROMI still outperforms other baselines on most datasets, and achieves the highest total score on both D4RL and NeoRL benchmarks**. We believe these results could validate the effectiveness of ROMI.
>
> **D4RL**
>
> | Dataset | CQL | IQL | MOPO | RAMBO | COUNT | MOBILE | ROMI |
> |-|-|-|-|-|-|-|-|
> | hopper-r | 8.2 $\pm$ 1.6 | 7.7 $\pm$ 0.3 | 29.5 $\pm$ 2.4 | 20.4 $\pm$ 6.2 | **31.7 $\pm$ 0.7** | 27.1 $\pm$ 4.0 | **32.1 $\pm$ 0.1** |
> | halfcheetah-r | 21.6 $\pm$ 2.8 | 16.7 $\pm$ 1.2 | 34.6 $\pm$ 2.2 | 32.6 $\pm$ 3.2 | 37.7 $\pm$ 2.4 | **41.3 $\pm$ 2.3** | **39.6 $\pm$ 3.3** |
> | walker2d-r | 5.4 $\pm$ 0.2 | 6.0 $\pm$ 0.7 | 0.0 | 0.0 | 22.4 $\pm$ 2.8 | **24.3 $\pm$ 1.9** | **25.8 $\pm$ 2.2** |
> | hopper-m | 59.7 $\pm$ 2.4 | 71.2 $\pm$ 2.8 | 76.4 $\pm$ 2.1 | 102.1 $\pm$ 3.6 | 105.6 $\pm$ 3.3 | 88.4 $\pm$ 6.9 | **113.5 $\pm$ 2.1** |
> | halfcheetah-m | 53.2 $\pm$ 2.5 | 49.6 $\pm$ 0.4 | 70.2 $\pm$ 3.6 | 80.0 $\pm$ 1.2 | 79.4 $\pm$ 1.5 | 78.8 $\pm$ 1.0 | **84.8 $\pm$ 1.0** |
> | walker2d-m | 78.9 $\pm$ 2.7 | 71.4 $\pm$ 1.6 | 86.6 $\pm$ 2.6 | 88.3 $\pm$ 4.6 | 93.5 $\pm$ 2.7 | 96.2 $\pm$ 2.7 | **104.3 $\pm$ 2.9** |
> | hopper-mr | 89.5 $\pm$ 2.1 | 98.3 $\pm$ 1.7 | 91.0 $\pm$ 3.2 | 81.1 $\pm$ 5.7 | 99.4 $\pm$ 3.9 | 93.0 $\pm$ 4.5 | **106.3 $\pm$ 3.5** |
> | halfcheetah-mr | 52.6 $\pm$ 0.7 | 48.3 $\pm$ 1.4 | 55.3 $\pm$ 1.3 | 72.1 $\pm$ 1.6 | 75.9 $\pm$ 2.6 | 71.3 $\pm$ 0.5 | **84.4 $\pm$ 1.3** |
> | walker2d-mr | 81.4 $\pm$ 2.1 | 77.0 $\pm$ 2.4 | 73.6 $\pm$ 1.5 | 81.8 $\pm$ 5.0 | **90.4 $\pm$ 1.4** | 80.0 $\pm$ 2.9 | 86.0 $\pm$ 1.4 |
> | hopper-me | 106.0 $\pm$ 2.5 | 100.2 $\pm$ 1.6 | 87.3 $\pm$ 2.0 | 103.4 $\pm$ 2.9 | **113.3 $\pm$ 4.8** | **114.3 $\pm$ 3.4** | **119.5 $\pm$ 3.8** |
> | halfcheetah-me | 80.3 $\pm$ 1.3 | 94.2 $\pm$ 1.6 | 93.6 $\pm$ 2.2 | 95.3 $\pm$ 2.3 | 103.2 $\pm$ 2.0 | **116.2 $\pm$ 3.0** | **117.2 $\pm$ 2.6** |
> | walker2d-me | 105.2 $\pm$ 0.2 | 111.0 $\pm$ 1.6 | 96.3 $\pm$ 2.9 | 77.6 $\pm$ 6.7 | 112.6 $\pm$ 0.4 | 104.1 $\pm$ 1.3 | **120.4 $\pm$ 0.8** |
> | Total | 742.0 | 751.6 | 794.4 | 834.7 | 965.1 | 935.0 | **1033.9** |
>
>
>
>
> **NeoRL**
>
>
> | Dataset | CQL | IQL | EDAC | MOPO | RAMBO | MOBILE | ROMI |
> |-|-|-|-|-|-|-|-|
> | Hopper-L | 17.4 $\pm$ 0.1 | 15.8 $\pm$ 1.2 | 14.7 $\pm$ 0.1 | 15.2 $\pm$ 0.7 | 17.4 $\pm$ 0.5 | 21.4 $\pm$ 1.3 | **26.7 $\pm$ 1.7** |
> | HalfCheetah-L | 35.4 $\pm$ 1.0 | 35.8 $\pm$ 0.5 | 32.6 $\pm$ 0.8 | 35.1 $\pm$ 2.8 | 33.6 $\pm$ 4.6 | **42.4 $\pm$ 3.8** | 39.0 $\pm$ 2.7 |
> | Walker2d-L | **43.7 $\pm$ 1.4** | 42.8 $\pm$ 2.0 | **44.0 $\pm$ 0.5** | 20.4 $\pm$ 2.1 | 26.2 $\pm$ 2.8 | 43.1 $\pm$ 1.9 | 41.4 $\pm$ 2.8 |
> | Hopper-M | **60.6 $\pm$ 2.7** | 58.2 $\pm$ 1.5 | 41.7 $\pm$ 2.9 | 0.0 | 44.5 $\pm$ 3.8 | 48.6 $\pm$ 6.7 | 53.7 $\pm$ 5.4 |
> | HalfCheetah-M | 55.0 $\pm$ 2.5 | 51.8 $\pm$ 0.1 | 57.6 $\pm$ 3.9 | 57.3 $\pm$ 1.9 | 54.3 $\pm$ 2.7 | 60.3 $\pm$ 2.5 | **66.8 $\pm$ 2.8** |
> | Walker2d-M | 54.3 $\pm$ 1.2 | 50.7 $\pm$ 3.9 | **56.4 $\pm$ 0.2** | 43.0 $\pm$ 2.1 | 38.8 $\pm$ 3.7 | 53.9 $\pm$ 2.7 | **58.2 $\pm$ 1.0** |
> | Hopper-H | **67.3 $\pm$ 3.0** | **65.0 $\pm$ 1.5** | 60.3 $\pm$ 1.7 | 15.6 $\pm$ 3.5 | 54.9 $\pm$ 5.1 | **65.5 $\pm$ 3.6** | **67.7 $\pm$ 2.9** |
> | HalfCheetah-H | 73.9 $\pm$ 0.3 | **79.6 $\pm$ 1.2** | 73.6 $\pm$ 1.4 | 68.0 $\pm$ 2.8 | 75.0 $\pm$ 2.5 | 70.8 $\pm$ 2.4 | **81.4 $\pm$ 3.2** |
> | Walker2d-H | 70.9 $\pm$ 0.8 | 73.7 $\pm$ 0.1 | **76.1 $\pm$ 1.6** | 30.2 $\pm$ 1.8 | 70.3 $\pm$ 3.9 | 71.2 $\pm$ 3.6 | **79.2 $\pm$ 1.2** |
> | Total | 478.5 | 473.4 | 457.0 | 284.8 | 415.0 | 477.2 | **514.1** |
>
> ## Concern 3: Larger compute requirement
>
> ROMI incurs only a slightly higher computational cost than RAMBO, and **the difference is not obvious**.
> As shown in **Table 5 of Appendix E.2**, the comparison of compute resource consumption between RAMBO and ROMI indicates that their parameter size, GPU memory usage, and runtime are comparable, with no significant differences. We believe that this slightly higher computational cost is justified, as ROMI explicitly addresses the issues of uncontrollable conservatism and unstable training, which are not handled by RAMBO.

---

> ### Author Response · Authors · 2025-11-23
> **Reply to Reviewer 7d1B (Part 3)**
>
> ## Concern 4: More hyperparameters are introduced
>
> First, we respectfully argue that **ROMI introduces only two additional hyperparameters**: $\xi$ and $N$. The rollout horizon $H$ is also used by other methods such as RAMBO, MOBILE, and Count-MORL, and is not unique to our work. Second, and more importantly, although new hyperparameters are introduced, ROMI eliminates the need for the highly sensitive adversarial coefficient $\lambda$ used in RAMBO, which causes over-conservatism and unstable updates. In contrast, $\xi$ provides precise control over the degree of conservatism, and $N$ is not a sensitive hyperparameter, as demonstrated in Table 7 of Appendix F.3. Thus, **these new hyperparameters contribute to more stable model updates and enable controllable conservatism**.
>
> ## Concern 5: Where are the results from for each method for D4RL and NeoRL benchmarks?
>
> We do not implement the baselines ourselves but use existing codebases for reproduction. As outlined in Appendix D.1, our main codebase is OfflineRL-kit [1], an open-source repository that provides implementations and tuned hyperparameters for CQL, IQL, EDAC, MOPO, RAMBO, MOBILE, etc. For Count-MORL, which is not included in OfflineRL-kit, we use its official codebase with the hyperparameters recommended in the original paper. We believe these baselines are correctly implemented and fairly tuned.
>
> ## Concern 6: How was Wasserstein distance chosen?
>
> Theoretically, not only the Wasserstein distance, but other distance metrics such as total variance distance or $f$-divergence could be used. However, **their dual reformulations often lead to complex constraints or regularizations** [2], and they are often limited to theoretical analysis and simple linear MDP settings. In contrast, **the Wasserstein distance admits an elegant closed-form dual reformulation (Proposition 4.1)**, which allows us to convert the dynamics uncertainty set into a simple state uncertainty set. **This property is critical for our practical implementation**. Therefore, we choose the Wasserstein distance instead of other distance metrics as our distance metric.
>
>
> [1] OfflineRL-Kit: An Elegant PyTorch Offline Reinforcement Learning Library. 2023
>
> [2] Distributionally robust stochastic programming. SIAM Journal on Optimization, 2017

---

> > ### Author Response · Authors · 2025-11-27
> >
> > Dear Reviewer 7d1B, we deeply appreciate your thoughtful review and your time, and hope that our response can address your concerns. We would like to kindly confirm if you still have any concerns or questions. We are more than happy to have further discussions with the reviewer if possible!

---

### Meta-Review · Area_Chair_SLBN · 2026-01-06

**Summary:**

ROMI addresses a real and practically important failure mode of RAMBO-style model-based offline RL—unstable training and hard-to-control conservatism—by introducing a robust value-aware model learning objective with an implicitly differentiable, adaptive weighting scheme (bi-level optimization). Across D4RL and NeoRL, and with additional 3M-step and Adroit results in the rebuttal, ROMI shows consistent gains over strong model-based baselines (RAMBO/MOBILE/COUNT), while keeping compute roughly comparable.

**Reviewer Concerns:**

Most concerns were addressed: the authors added 3M-step comparisons showing ROMI remains competitive against baselines tuned longer; provided compute/memory/runtime comparisons indicating only marginal overhead versus RAMBO; clarified baseline implementations come from established codebases (OfflineRL-Kit / official code) and reported full uncertainty intervals in the revision/appendix; expanded evaluation beyond narrow MuJoCo by adding Adroit, and clarified why Wasserstein distance is used (tractable dual form enabling a practical uncertainty set). Remaining issues (statistical reporting style and the broader community practice around offline hyperparameter tuning) are valid but not specific blockers for this submission.

**Reviewer Scores:**

The panel skews positive overall, with two clear accepts (including a detailed “no major issues” accept) and one borderline review whose main fairness and training-length concerns were directly answered with new 3M-step results and fuller reporting. Given the additional experiments and clarifications, the likely end-state is a stable accept consensus with the main requested changes being presentation/rigor improvements rather than substantive technical gaps.

---

### Decision · Program_Chairs · 2026-01-26

Accept (Poster)